# Design of stretchable and self-powered sensing device for portable and remote trace biomarkers detection

Wenxi Huang[1,5], Qiongling Ding[1,5], Hao Wang[1,5], Zixuan Wu[1], Yibing Luo[1], Wenxiong Shi[2], Le Yang[3,4], Yujie Liang [3,4], Chuan Liu[1] & Jin Wu [1] ✉

Timely and remote biomarker detection is highly desired in personalized medicine and health protection but presents great challenges in the devices reported so far. Here, we present a cost-effective, flexible and self-powered sensing device for $H_2S$ biomarker analysis in various application scenarios based on the structure of galvanic cells. The sensing mechanism is attributed to the change in electrode potential resulting from the chemical adsorption of gas molecules on the electrode surfaces. Intrinsically stretchable organohydrogels are used as solid-state electrolytes to enable stable and long-term operation of devices under stretching deformation or in various environments. The resulting open-circuit sensing device exhibits high sensitivity, low detection limit, and excellent selectivity for $H_2S$. Its application in the non-invasive halitosis diagnosis and identification of meat spoilage is demonstrated, emerging great commercial value in portable medical electronics and food security. A wireless sensory system has also been developed for remote $H_2S$ monitoring with the participation of Bluetooth and cloud technologies. This work breaks through the shortcomings in the traditional chemiresistive sensors, offering a direction and theoretical foundation for designing wearable sensors catering to other stimulus detection requirements.

Over the past few decades, biomarker detection has garnered considerable attention in the medical and health domain due to its potential for early disease diagnosis and personalized health monitoring, and it is expected to play a crucial role in the forthcoming era of cloud medical treatment (CMT)[1,2]. Extensive studies have shown that there are more than 800 compounds in human exhaled breath, some of which are closely related to the metabolism of different organs, with different levels between patients and healthy people, making them ideal in vitro biomarkers for non-invasive diagnostics[3]. Wherein, hydrogen sulfide ($H_2S$) is a typical biomarker caused by oral diseases

such as tongue coating, periodontal disease, and mucosal disease, produced by the metabolism of organic matter by accumulated bacteria, which bothers millions of people[4,5]. In a clinical study, the average $H_2S$ concentration in mouth breath of patients with oral halitosis is 20.64 ppb, much higher than the value of healthy volunteers (3.36 ppb)[6,7], which indicates that $H_2S$ can be utilized as a judgment index for non-invasive halitosis diagnosis. Similarly, $H_2S$ can also be employed as a biomarker for the early identification of meat spoilage after the growth of bacteria[8]. Therefore, the accurate monitoring of ppb-level $H_2S$ is highly desirable for diverse applications spanning

[1]State Key Laboratory of Optoelectronic Materials and Technologies, Guangdong Province Key Laboratory of Display Material and Technology, School of Electronics and Information Technology, Sun Yat-sen University, 510275 Guangzhou, China. [2]Institute for New Energy Materials and Low Carbon Technologies, School of Materials Science and Engineering, Tianjin University of Technology, 300384 Tianjin, China. [3]Department of Oral and Maxillofacial Surgery, Guanghua School of Stomatology, Hospital of Stomatology, Sun Yat-sen University, 56th Lingyuanxi Road, 510055 Guangzhou, Guangdong, China. [4]Guangdong Province Key Laboratory of Stomatology, No. 74, 2nd Zhongshan Road, 510080 Guangzhou, Guangdong, China. [5]These authors contributed equally: Wenxi Huang, Qiongling Ding, Hao Wang. ✉e-mail: wujin8@mail.sysu.edu.cn

from medical health to food safety. Besides, continuous monitoring of relatively high concentrations of $H_2S$ is imperative due to its toxic nature. Prolonged exposure to $H_2S$ levels exceeding 10 ppm can have severe detrimental effects on human health[9]. Currently, methods for accurate detection of $H_2S$ in medical diagnosis include gas chromatography/mass spectrometry[10], ion chromatography[7], etc. However, these methods are cumbersome, costly, and time-consuming, failing to achieve real-time detection. Therefore, the development of portable gas sensors capable of detecting low concentrations of $H_2S$ is urgently needed.

So far, mainstream research efforts have demonstrated a series of chemoresistive gas sensors based on metal oxide semiconductors (MOSs), graphene, and two-dimensional transition-metal dichalcogenides (TMDs) for timely $H_2S$ detection[11–13]. Nevertheless, these devices always present a dilemma between sensing performance and operating temperature and the development of highly sensitive $H_2S$ sensors at room temperature (RT) is particularly challenging. For example, Yun et al. decorated the Cu-doped ZnO nanostructures with reduced graphene oxide (RGO) for $H_2S$ gas detection[14]. Although it can detect $H_2S$ in a wide range of 3–10 ppm at RT, its theoretical limit of detection (LOD) is only about 136 ppb, which is far from meeting the performance requirements of biomarker detection. Li et al. fabricated an $\alpha$-$Fe_2O_3$ nanoparticle-based $H_2S$ sensor with a LOD of 50 ppb, but an operating temperature of 300 °C was required[15]. Such a high operating temperature not only poses serious safety concerns but also greatly increases power consumption, which is difficult to apply in portable and wearable electronics. For this type of sensor, $H_2S$ can only be adsorbed weakly on the surface of the sensing material at low temperatures, which is insufficient to cause a large resistance change in the overall material, thus resulting in suboptimal sensing performance. In actual commercialization, electrochemical gas sensors are the most dominant type for RT $H_2S$ detection, with a broad linear detection range. Its sensing mechanism is derived from the Faradaic current generated by the electrochemical oxidation of $H_2S$ on the working electrode, which requires the continuous provision of appropriate reaction potential and well-designed highly catalytic electrode materials, giving rise to high power consumption and cost. Taking into account the easy leakage and volatilization of the commonly used electrolyte solution, such devices are always delicately packaged and exhibit large volume, rigidity, and limited lifetime, making them challenging to proceed with further miniaturization and be applied in wearable electronics.

To adapt to the demands of wearable electronics, the most important thing is to achieve flexibility and even stretchability in gas sensors, allowing them to conform to clothing or skin[16]. One strategy to achieve flexible gas sensors is integrating sensing materials on soft substrates (e.g., polyethylene terephthalate, polydimethylsiloxane, polyimide, and Ecoflex)[17–22]. However, the stretchability of sensors obtained by the above method was highly limited by the substrates, and the rigid sensing materials will be damaged after being subjected to external mechanical stress. Considering the inevitable deformation, wear, and collision of sensor devices during human motion, another more effective strategy has received great attention by developing intrinsically stretchable sensing materials[23–25]. Being composed of water and three-dimensional polymeric networks, hydrogels have been shown to be one of the most potential sensing components in wearable applications in recent years, thanks to their intrinsic stretchability, transparency, biocompatibility, and stimulus-responsiveness[26–30]. In fact, hydrogel-based sensors that are sensitive to stimuli including strain, pressure, temperature, and humidity have burst forth in the scientific community over the past decade[31–34]. Nevertheless, research on hydrogel-based gas sensors remains relatively infrequent, and the development of hydrogel-based $H_2S$ sensors has been especially limited thus far.

Another long-standing challenge for wearable or portable electronics is the need for a continuous external power supply. Driven by this, several self-powered sensing devices based on various energy harvesting technologies have emerged, including triboelectricity, piezoelectricity, thermoelectricity, optoelectronics, etc[35–37]. Whereas, these techniques are difficult to use in the measurement of static ambient gases since they always require specific energy sources (motion, heat, or light) to drive, and the sensing performances are also easily disturbed by them. To this end, the exploration of spontaneous self-powered gas sensors for continuous gas monitoring is highly desirable. Galvanic cells are ideal spontaneous discharge devices that can be used for gas detection due to electrochemical reactions, but they are accompanied by severe electrode corrosion at the negative electrode and exhibit a short device lifetime. Focusing on the electrode, although the limited target gas adsorption on its surface at RT is difficult to change the conductance of the bulk electrode, it is very likely to affect its electrode potential in relation to the electrode surface state in contact with the electrolyte. Thus, it is expected that a self-powered and sustainable gas sensor coupled with a galvanic cell can be developed by real-time testing of its open-circuit voltage (OCV), which has never been addressed.

Inspired by this, herein, a self-powered and stretchable $H_2S$ sensor is proposed with a galvanic cell-like structure consisting of two different metal electrodes and a polyacrylamide (PAM)/calcium alginate (CA) double-network (DN) hydrogel. Wherein, the water-rich and ion-conducting PAM/CA hydrogel serves as the solid-state and stretchable electrolyte in the system, endowing the sensor with high flexibility and stretchability. When exposed to $H_2S$, one of the metal electrodes is used as the active electrode to interact with the target gas, and the other metal electrode is used as a reference. The OCV of the device is monitored to represent the electrode potential difference between the two metal electrodes on the hydrogel surface, no external power supply is required and no rapid electrode corrosion occurs. After exploration, the effects of $H_2S$ on various metal electrodes show obvious differences, and its chemical adsorption on Ag electrodes is the strongest, which has good reversibility when $H_2S$ is removed. As a result, the fabricated gas sensor exhibited superior sensitivity, repeatability, and excellent selectivity at RT (25 °C), competent in working under severe mechanical deformations or in aerobic/anaerobic environments, favoring practical application development. After a solvent replacement with glycerol (Gly) in the hydrogel, the stability and environmental tolerance of the sensing device is greatly improved accordingly. Besides, the LOD of the $H_2S$ sensor is calculated to be 0.79 ppb, which is lower than that of state-of-the-art $H_2S$ sensors, so it is possible to detect $H_2S$ biomarkers released by bacteria. To demonstrate this, both halitosis diagnosis and meat spoilage monitoring are carried out by using the developed $H_2S$ sensor, enabling real-time identification of halitosis sufferers as well as spoiled pork. Finally, a wireless $H_2S$ alarm system that can identify $H_2S$ leaks in real-time and remotely through Bluetooth or data cloud sharing has also been developed, illustrating its application potential in health and safety protection and the Internet of Things.

## Results
### Preparation and characterization of hydrogel electrolyte
The highly stretchable and tough PAM/CA DN hydrogel (DNH) electrolyte was synthesized via a facile, two-step strategy (Experimental section and Fig. 1a). From the scanning electron microscope (SEM) image of the freeze-dried DNH, the polymer components in the hydrogel appeared as a uniform interpenetrating porous structure in which water was filled (Supplementary Fig. 1), enabling easy mass transfer as in liquid electrolytes. As shown in Supplementary Fig. 2 and Supplementary Movie 1, the hydrogel was able to be easily stretched up to 400% strain and remained intact and undamaged due to the complementary mechanical properties of the two polymer networks involved[38]. Concretely speaking, during stretching, the weaker physically crosslinked CA network would be easier to unravel to dissipate

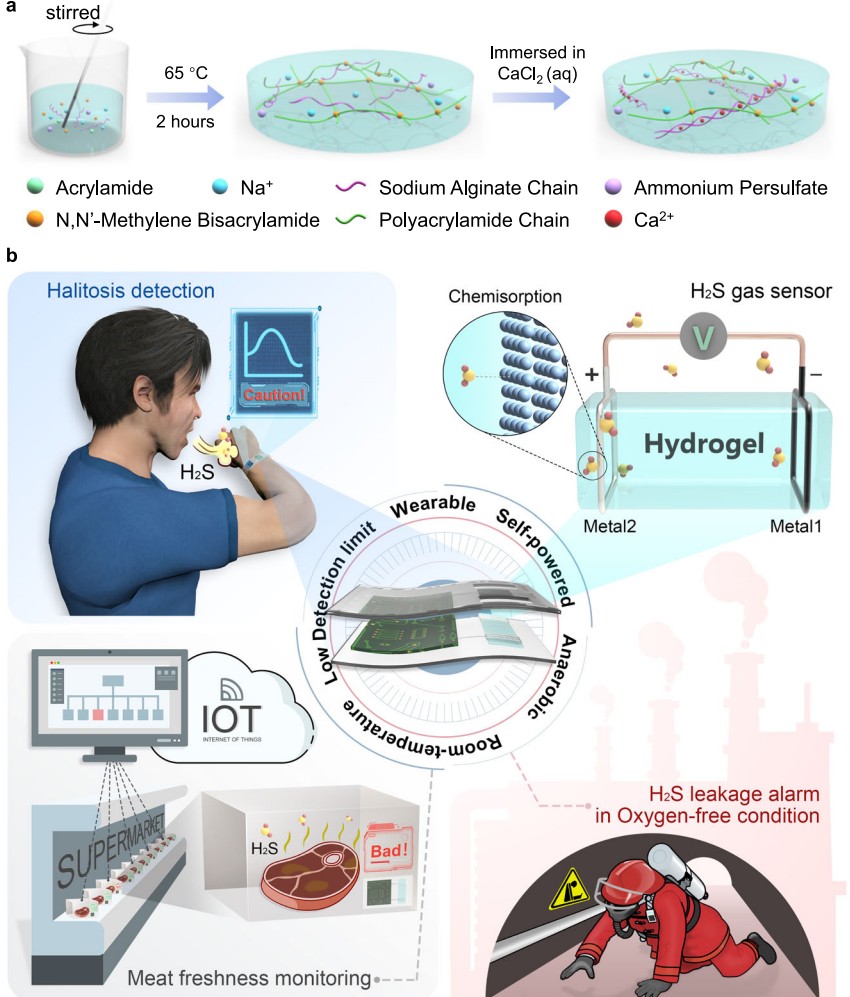

**Fig. 1 | Schematic illustration for the preparation and application of Zn/Ag/ DNH sensor. a** Schematic illustrating the fabrication process of PAM/CA DN hydrogel. **b** Schematic illustration showing the flexible gas sensing device and its application in halitosis detection, meat freshness monitoring, and H₂S leakage alarm.

energy, and the tougher chemically cross-linked PAM network would remain unaffected thereby maintaining the integrity of the system. Attractively, the DN hydrogel exhibited remarkable optical transparency, and the transmittance to visible light was higher than 83% (Supplementary Fig. 3). When the prepared hydrogel was placed on paper, the image of H₂S on the paper could be seen clearly through it. Benefiting from this, the hydrogel may even be used in the construction of invisible devices and further applied in military camouflage[39], which is a unique advantage over other kinds of sensors. Unfortunately, hydrogels always have poor frost and desiccation resistance due to the presence of a large amount of free water, and their application scenarios are limited. To address this, a solvent replacement strategy was adopted to convert pristine hydrogels to organohydrogels, where the Gly molecules infiltrated and exchanged with the water molecules in the pristine hydrogels by immersing them in pure Gly for 1, 2, or 4 h[40]. The obtained DN organohydrogels were named Gly1h-DNO, Gly2h-DNO, and Gly4h-DNO, respectively. By placing the pristine DN hydrogel and organohydrogels in a relatively dry environment (25 °C, 40% relative humidity (RH)) for 36 h, their morphology and mass were recorded at intervals to evaluate the moisture retention ability (Supplementary Figs. 4, 5). For the pristine DN hydrogel, it shrank rapidly and lost 72% of its initial weight within 10 h, showing a poor anti-drying ability. After the introduction of Gly, the appearance changes of the samples were greatly reduced, and their moisture

retention ability increased with the increase of soaking time. Within 10 h, the mass loss of Gly1h-DNO, Gly2h-DNO, and Gly4h-DNO was greatly reduced to 38.2%, 31.7%, and 18.9%, respectively, demonstrating enhanced moisturizing ability. Also, according to the differential scanning calorimetry (DSC) tests of the pristine DN hydrogel and organohydrogels, the freezing resistance was also greatly enhanced when Gly was introduced. As shown in Supplementary Fig. 6, the freezing point of the Gly1h-DNO decreased to −42.4 °C compared to the pristine DN hydrogel (−18.6 °C), and further Gly2h-DNO and Gly4h-DNO even lowered the freezing point to below −120 °C since no peak was observed in their DSC curves.

Fundamentally, this enhanced environmental stability can be attributed to the greatly reduced free water content in the hydrogel system after the introduction of Gly. Specifically, there are three hydroxyls on a Gly molecule, and these hydroxyl groups can bond with surrounding water molecules through hydrogen bonds to reduce the energy of the system, making most of the free water molecules bound in the network. To confirm this, the Fourier transform-infrared (FTIR) spectra of the pristine hydrogel and organohydrogel were recorded in Supplementary Fig. 7. In terms of pristine hydrogel, the C=O peak at 1670 cm⁻¹ as well as the N−H peaks at 1620 and 3200 cm⁻¹ came from the carboxyl groups on CA and the amino groups on PAM, respectively, indicating the coexistence of PAM and CA molecular chains. Besides, the peak at

3400 cm$^{-1}$ was associated with the O–H bonds originating from the hydroxyls on both polymer networks. As for the organohydrogels, it could be found that the peak of the hydroxyl group was significantly increased due to the addition of Gly and the water molecules could be bound by these hydroxyl groups through hydrogen bonds, thus greatly inhibiting the evaporation and freezing of water molecules in the hydrogel. In addition, we also observed a decrease in C=O peak and a corresponding increase in C–O peak, and it could be inferred that some C=O was converted to C-O through bonding with hydroxyl groups on the Gly molecules. Considering the rapidly increasing number of hydrogen bonds in the network system, this would also lead to an increase in the mechanical strength of the organohydrogel due to the increase in crosslink density. As shown in Supplementary Fig. 8a, the organohydrogels, along with the pristine hydrogels, were subjected to incremental tensile strain until they broke to gain the stress-strain curves. Extracted from the linear region of these stress–strain curves, Young's moduli of pristine hydrogel, Gly1h-DNO, Gly2h-DNO, and Gly4h-DNO were determined as 77.5, 149, 182, and 252 kPa, respectively (Supplementary Fig. 8b), reflecting the better mechanical strength in organohydrogels. Thanks to the good biocompatibility and non-flammability of the constituent materials as well as the good mechanical strength, stretchability, and environmental stability of the system, organohydrogels can work flexibly in various open environments without safety hazards and have great advantages over conventional liquid electrolyte solutions.

## Design and optimization of device structures

Here, a self-powered H$_2$S sensor inspired by galvanic cells was assembled with a typical metal electrode–electrolyte–metal electrode structure, and it could be used as a revolutionary wearable device for the monitoring of H$_2$S biomarkers in practice (Fig. 1b). The developed solid-state hydrogel is utilized as an electrolyte, which enables the device to withstand various deformations and avoid leakage and combustion. In addition, some common metal wires are used to serve as two electrodes without other modifications, possessing highly low cost as well as good mechanical strength and flexibility. It is widely recognized that metal lattices comprise of systematically arranged metal ions and freely mobile electrons. When a metal electrode is submerged in water, the strongly polar water molecules tend to attract metal ions located within the metal lattices, thereby weakening the bond between certain metal ions and others present in the metal structure. The metal ions with weakened bonds then dissolve into the water, leading to a negative charge on the metal electrode due to the loss of metal ions. Once the dissolution and precipitation of metal ions reach a dynamic equilibrium, the metal electrode demonstrates a stable electrode potential. Different metal electrodes display distinctive electrode potentials in the electrolyte, owing to their unique properties. Based on this principle, a self-powered H$_2$S sensing device with a steady OCV is manufactured utilizing two different metals as electrodes. For the convenience of presentation, the fabricated devices are declared as metal 1/metal 2/electrolyte (e.g., Zn/Ag/DNH), where metal 1 connects the negative electrode and metal 2 connects the positive electrode of the test instrument, respectively (Supplementary Fig. 9). Then, the OCV of sensing devices that consisted of DNH and different metal electrodes were recorded, including Zn/Ag/DNH, Zn/Cu/DNH, Fe/Ag/DNH, Fe/Cu/DNH, and Cu/Ag/DNH (Supplementary Fig. 10). Among them, the Zn/Ag/DNH showed a high and stable OCV of 949.8 mV, indicating the self-powering and sensing capacity of the prepared devices.

When the sensors are used for gas detection, the OCV can show a tight correlation with the H$_2$S concentration. As shown in Supplementary Fig. 11, the OCV of the Zn/Ag/DNH dropped sharply when

exposed to H$_2$S and then recovered in N$_2$ gas, which is fully capable of being used for H$_2$S sensing. The response here is defined as

$$Resp = \Delta V = V_{H_2S} - V_0 \qquad (1)$$

where $V_{H_2S}$ and $V_0$ are the stabilized OCV in flowing H$_2$S and background gas, respectively. Notably, the measurement of OCV necessitates no external power source throughout the process, so the fabricated devices are self-powered.

The H$_2$S-sensing mechanism of this spontaneous self-powered device can be attributed to the reversible chemisorption of the target gas by the electrodes (Fig. 2a). Theoretically, the electrode potential of an electrode is mainly related to the electrode components in contact with the electrolyte, and the chemical adsorption or reaction of external disturbances on the electrode will inevitably lead to changes in the electrode potential, thereby causing variations in OCV. Considering the irreversibility of the chemical reaction, H$_2$S is only weakly bonded to the electrode in our case, and the charge transfer exists. In order to explore the influence of electrodes on the gas-sensing performance, sensors assembled from some common metal wires were compared, including Zn, Ag, Fe, and Cu wires, and the prepared DNH was used as the electrolyte. When exposed to H$_2$S, the dynamic response curves of Zn/Ag/DNH, Fe/Ag/DNH, and Zn/Fe/DNH were shown in Fig. 2b. Prior to this, a baseline correction had been carried out to exclude the influence of baseline offset caused by slow electrode recovery, the details of which were described in Supplementary Fig. 12. Apparently, both Zn/Ag/DNH and Fe/Ag/DNH displayed significant changes in OCV and increased responses with increasing H$_2$S concentration, while Zn/Fe/DNH barely showed any change in OCV. This suggests that the key to the response lies in the employment of the Ag electrode, which can interact with H$_2$S to cause a change in the surface composition, leading to a decrease in the Ag electrode potential and subsequently the OCV of the device. And Zn and Fe have little interaction with H$_2$S, thus leading to the insensitivity of Zn/Fe/DNH to H$_2$S. Although H$_2$S can definitely be adsorbed and dissociated on the surface of hydrogel due to its hydrophilicity, this is obviously not enough to have an effect on the electrode potential of the metal wire according to experimental phenomena. After H$_2$S was removed, the OCV of the device gradually recovered, indicating the weak reversible chemisorption of H$_2$S on the electrode rather than chemical reactions. Due to the existence of the concentration gradient, the adsorbed H$_2$S molecules can break free and diffuse into the dynamic airflow and be taken away, resulting in the restoration of the Ag electrode potential.

According to further exploration, Zn/Cu/DNH and Fe/Cu/DNH could also be used for H$_2$S detection, but with small responses, implying weaker chemisorption of H$_2$S on Cu electrodes compared to Ag electrodes (Supplementary Fig. 13). Considering the rapid corrosion rate of Fe in humid environments, Zn and Ag electrodes were chosen for further testing. As discussed earlier, hydrogels exhibit poor moisture retention and can lead to device failure when dried out. Therefore, the organohydrogel containing water/Gly binary solvent can significantly prolong the lifetime of devices, and the effect of Gly content on gas sensing performance was studied. As shown in Fig. 2c, the dynamic response curves of devices assembled from hydrogels soaked in Gly for different times (0, 1, 2, and 4 h) were tested. It could be clearly seen that the response decreased when part of the water in the hydrogel was replaced by Gly and the response of Gly4h-DNO was the smallest. It is inferred that the reduction of water molecules in the hydrogel would be detrimental to the chemisorption of H$_2$S molecules on the electrodes at the interface. To balance the stability and responsiveness of the device, Gly1h-DNO was chosen for the construction of the sensor and directly referred to as DNO later. During the long-term test (96 h), the OCV of both Zn/Ag/DNH and Zn/Ag/DNO sensors remained basically stable (Supplementary Fig. 14), showing the

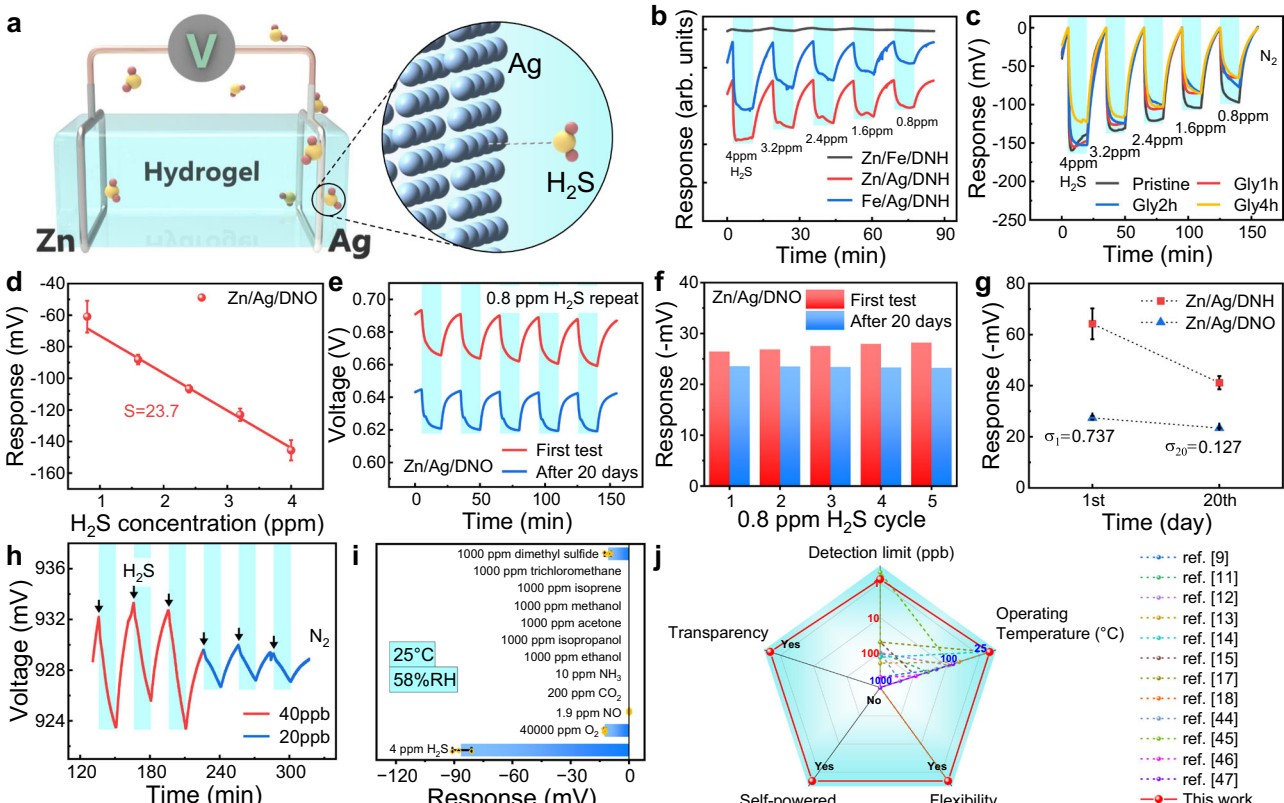

**Fig. 2 | H₂S sensing properties of Zn/Ag/DNO. a** Schematic diagram illustrating the device structure and H₂S-sensing mechanism of Zn/Ag/DNO gas sensor. **b** Dynamic responses of Zn/Fe/DNH, Zn/Ag/DNH, and Fe/Ag/DNH to H₂S gas with reduced concentration from 4 to 0.8 ppm. **c** Dynamic responses of Zn/Ag/DNH and Zn/Ag/DNOs to H₂S gas with reduced concentration from 4 to 0.8 ppm after baseline corrections. **d** Average response (dots) of three Zn/Ag/Gly1h-DNO samples versus H₂S concentration and corresponding linear fitting line that revealed the sensitivity. The error bars denote standard deviations of the mean. **e** Dynamic OCV and **f** response histogram of Zn/Ag/DNO sensor to 0.8 ppm H₂S cycling for the first time and after 20 days of placement. **g** Comparison of Zn/Ag/DNH and Zn/Ag/DNO sensors' responses to 0.8 ppm H₂S from the first test and the test after 20 days of placement. $n = 5$ for each group. The error bars denote the standard deviations ($\sigma$) of the mean. **h** Experimental detection limit of Zn/Ag/DNO sensor to H₂S. 20 ppb is the minimum concentration that can be achieved under current experimental conditions. **i** Comparison in the responses of the Zn/Ag/DNO sensors to various gaseous chemicals. $n = 3$ for the H₂S group and $n = 5$ for the other groups. The error bars denote standard deviations of the mean. The precise mean responses are −86.56, −12.47, −0.13, and −10.69 mV for H₂S, O₂, NO, and dimethyl sulfide, respectively. Responses to the other interfering chemicals are imperceptible. **j** Capability radar comparing the performance of the state-of-the-art H₂S sensors.

stability of the devices. During the sensor construction, metal wires were wound on both ends of the gel as electrodes, and the contact area between the electrodes and the hydrogel may also affect the sensing performance, which was then qualitatively studied by varying the number of turns and diameter of the Ag wires. From Supplementary Figs. 15, 16, these devices have a similar response to H₂S, and the contact area of the Ag wire on the gel has little effect on the sensing performance due to the uniform adsorption per unit area. It can be considered to further increase the responsivity of the device by elevating the adsorption sites of the active electrode, which can be followed up.

### Sensing performance of Zn/Ag/DNOs

Sensitivity is an important performance index of the gas sensor, which refers to the variation of a response brought on by a unit of gas concentration and can be estimated from the slope of the response versus gas concentration curve:

$$S = \left| \frac{\Delta Resp}{\Delta C} \right| \qquad (2)$$

For three constructed Zn/Ag/DNO sensors, their average response versus H₂S concentration curve was shown in Fig. 2d according to its dynamic response curve towards different H₂S concentrations ranging

from 4 to 0.8 ppm. The results show that there is a good linear relationship between the response and the H₂S concentration, which is beneficial for the practical resolution of the H₂S concentration. Besides, the small standard deviations in the response of the three Zn/Ag/DNO samples indicate excellent reproducibility. Based on the corresponding linear fitting curve, the sensitivity of the Zn/Ag/DNO sensor to H₂S was determined to be 23.7 mV/ppm, fully demonstrating its capability in detecting H₂S. Nevertheless, the sensor exhibited a moderate response speed, which can be considered as an area for improvement (Supplementary Fig. 17). Further, the repeatability and stability of the device were investigated, which are also important parameters for gas sensors in practical applications. By repeatedly exposing the Zn/Ag/DNO sensor to 0.8 ppm H₂S for five times, its dynamic OCV curve was recorded, as presented in Fig. 2e. Apparently, the OCV variation of the devices to the same H₂S concentration is almost consistent, with an average of 27.4 mV and a standard deviation ($\sigma$) of 0.737 (Fig. 2f), indicating its good repeatability. After 20 days of placement in the ambient environment, the repeatability test of the same device to 0.8 ppm H₂S was carried out and its performance was compared to that before placement. It could be found that the device still exhibits good repeatability, and the response to 0.8 ppm H₂S only drops by 4 mV and reaches 23.4 mV with a standard deviation of 0.127, demonstrating good stability. Note that the initial OCV of the device was decreased due to the electrode potential shift caused by electrode

oxidation that occurs during placement. In stark contrast to the hydrogel-based sensor, its response dropped by about one-third after 20 days, with poor stability (Supplementary Fig. 18 and Fig. 2g). Even though the response of Zn/Ag/DNH sensor to 0.8 ppm $H_2S$ initially reaches around 64.21, it is not suitable for practical applications due to its unstable responsiveness. The rapid water loss in DNH may be responsible for the fall in response, whereas Gly1h-DNO performed better at moisturizing and had less of a decline.

For biomarker detection, the gas sensors must be able to react to ppb-level $H_2S$. Here, we subjected the Zn/Ag/DNO sensor to a lower-concentration $H_2S$ to evaluate its LOD (Supplementary Fig. 19 and Fig. 2h). Notably, the sensor showed a distinguishable response to 20 ppb $H_2S$, which is the lowest concentration of $H_2S$ we can obtain limited by the current experimental conditions and far lower than most of the state-of-the-art $H_2S$ gas sensors. Based on the sensitivity and the noise level, the theoretical LOD can be calculated as 0.79 ppb, which is enough to tell if the test subject has a bad breath problem or a tendency to halitosis (Supplementary Fig. 20 and Supplementary Table 1)[41-43]. Selectivity is one of the most important performance indicators of a gas sensor, referring to the ability to identify target gases from various interfering gases, which was investigated by comparing the response to $H_2S$ and those to various interference gases. In this case, we investigated the response of the Zn/Ag/DNO sensor to some possible interfering gases in the environment or in exhaled air at 25 °C, 58% RH, including 40,000 ppm $O_2$, 1.9 ppm NO, 200 ppm $CO_2$, 10 ppm $NH_3$, 1000 ppm ethanol, 1000 ppm isopropanol, 1000 ppm acetone, 1000 ppm methanol, 1000 ppm isoprene, 1000 ppm trichloromethane, and 1000 ppm dimethyl sulfide (Supplementary Fig. 21). Although 40,000 ppm $O_2$ reduced the OCV of the sensor by 12.47 mV, it had little effect on the detection of $H_2S$ when the $O_2$ concentration changed little. Furthermore, the sensor demonstrated a response of −10.69 mV to 1000 ppm dimethyl sulfide, implying its preference for sulfur-containing gases. Except for $O_2$ and dimethyl sulfide, the gas sensor exhibited a negligible response of 0.13 mV to 1.9 ppm NO and imperceptible responses to other interfering chemicals, demonstrating excellent selectivity (Fig. 2i). And this could be attributed to the sulfurophilic nature of Ag and the promoting effect of $H_2S$ on the dissolution of metal ions from the Ag electrode, which was further validated by the electrochemical impedance spectroscopy (EIS) of the sensor under different gas atmospheres (Supplementary Fig. 22). Combining the advantages of spontaneous self-powered ability, low LOD, room-temperature operation, high stretchability, and transparency, the Zn/Ag/DNO sensor is more suitable for wearable device application than the others (Supplementary Table 2 and Fig. 2j)[9,11-15,17,18,44-47].

## Environmental compatibility

Previously reported gas sensors often have quite poor anti-interference ability, and some common interference factors such as increased humidity, lowered temperature, deformation, and oxygen scarcity may cause the sensing performance of the device to seriously decline or even fail, which greatly restricts their practical application. Satisfactorily, the environmental inclusiveness of the sensor we developed has been greatly expanded, and it can work normally under various environmental conditions to adapt to different application scenarios. Firstly, the effect of humidity on the gas sensing performance of the Zn/Ag/DNO sensor was explored. As shown in Fig. 3a, the dynamic response curves of the Zn/Ag/DNO sensor to 4−0.8 ppm $H_2S$ in different humidity environments (37%, 46%, 58%, and 80% RH) were obtained. With the increase of RH, the sensor responsivity increased gradually, from 13.6 mV at 37% RH to 55.7 mV at 80% RH for 0.8 ppm $H_2S$ (Fig. 3b). The sensor exhibited greater response at 80% RH, and this can be explained that more $H_2S$ molecules could be adsorbed and further reacted on the wetted electrode due to its hydrophilic nature, resulting in a larger response to the same $H_2S$ concentration. While in

the absence of water molecules, $H_2S$ is difficult to adsorb directly on the bare Ag electrode. In our case, despite the low responsivity, our sensor can perform $H_2S$ sensing even in a dry environment thanks to the presence of water in the hydrogel, enabling $H_2S$ detection in a wide humidity range. And the influence of humidity on the response value can be further eliminated by encapsulation with hydrophobic and breathable elastomeric membranes (Supplementary Fig. 23). Then, the operating temperature range of the sensor was investigated. We measured the $H_2S$-sensing performance of the Zn/Ag/DNO sensor at different temperatures from −20 to 40 °C (Fig. 3c, d), a temperature range that covers a large part of daily life. The gas sensing performance of the sensor at −20 and 40 °C exhibits similar characteristics, with sensitivities of 7.8 and 6.3 mV/ppm, and detection limits of 3.37 and 3.98 ppb, respectively. Despite the reduced response compared to RT, it retains the capability to detect $H_2S$, thus satisfying the demands for detecting $H_2S$ leakage in certain challenging operational conditions. To eliminate the interference of temperature, a temperature sensor can be employed to accurately measure the ambient temperature, and the gas sensor can subsequently be calibrated based on the response curve obtained at various temperatures. Thanks to the feasibility of normal operation under various environmental conditions, our sensors are expected to be used in various application scenarios, including hot summers, cold winters, wet rainy seasons, and arid deserts.

After that, the effect of mechanical deformation on the sensing performance of the sensor was also investigated. Owing to the excellent flexibility and stretchability of the PAM/CA organohydrogel, the Zn/Ag/DNO sensor can be easily stretched without damage. As demonstrated in Fig. 3e, after the sensors were stretched to different strains (0%, 50%, and 100%), their dynamic response curves to $H_2S$ from 4 to 0.8 ppm were tested. It can be found that the responsiveness of the sensor is enhanced under tensile strain (Fig. 3f, g). The difference in response between the stretched and original states of the sensor can be attributed to changes in the interface, which can be addressed by further refining the structural design and implementing appropriate encapsulation techniques. The Zn/Ag/DNO sensor's ability to function effectively under strain makes it an ideal candidate for wearable electronics. With regard to traditional gas sensors based on MOSs, their response strongly depends on the generation of adsorbed oxygen and cannot be used in anaerobic environments. Finally, unlike these devices, the Zn/Ag/DNO sensor can detect $H_2S$ in both aerobic and anaerobic environments. Here, we used air as the background gas and the balance gas of $H_2S$ to test the gas sensing performance, which was then compared with that of the situation using $N_2$ as the background gas and balance gas (Fig. 3h). As we can see from Fig. 3i, the sensor exhibited a sensitivity of 12.1 mV/ppm and a LOD of 1.66 ppb to $H_2S$ in aerobic environments. The better sensing performance of the sensor in an anaerobic environment is attributed to the oxygen-independent sensing mechanism. When oxygen is present, it can compete with $H_2S$ for adsorption sites, giving rise to a decrease in response. As a consequence, thanks to the $H_2S$ sensing capability in both air and $N_2$ atmospheres, the application of the Zn/Ag/DNO sensor in oxygen-deficient environments like mines and in aerobic environments such as tanneries and refineries is possible, not restricted by the oxygen concentration.

## Sensing mechanism

As for the traditional galvanic gas sensor, the sensing mechanism is ascribed to the faradic current generated by the electrochemical reaction of the target gas at the electrode. Whereas, such devices invariably suffer severe corrosion at the negative electrode during operation, which greatly impairs their service life. Currently, they are limited to the detection of electrochemically active oxidizing gases. With respect to the spontaneous self-powered Zn/Ag/DNO sensor developed in our case, it can perform highly sensitive, reproducible, and stable detection of reduced $H_2S$ under various environmental

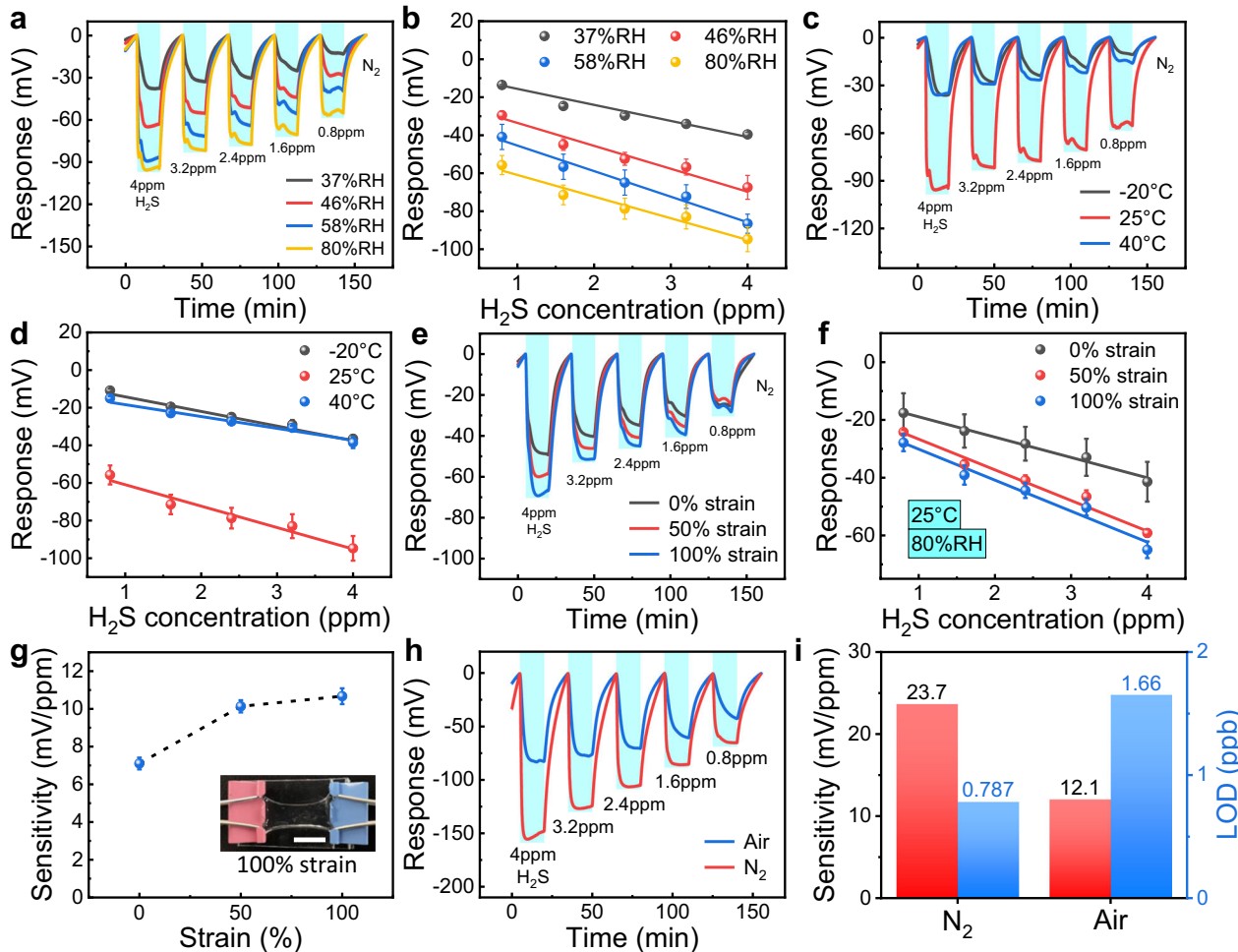

**Fig. 3 | Sensing performance under variable environmental conditions and scenarios. a** Dynamic response of the Zn/Ag/DNO sensor to $H_2S$ gas with reduced concentration from 4 to 0.8 ppm under different RH. **b** Response versus concentration curves of the Zn/Ag/DNO sensor at different RH. $n = 3$ for each group. The error bars denote standard deviations of the mean. **c** Dynamic responses of the Zn/Ag/DNO sensor to $H_2S$ gas with reduced concentration from 4 to 0.8 ppm under different temperatures. **d** Response versus concentration curves of the Zn/Ag/DNO sensor under different temperatures. $n = 3$ for each group. The error bars denote standard deviations of the mean. **e** Dynamic responses of the Zn/Ag/DNO sensor to $H_2S$ gas with reduced concentration from 4 to 0.8 ppm under different external strains. **f** Response versus concentration curves of the Zn/Ag/DNO sensor under different external strains. $n = 3$ for each group. The error bars denote standard deviations of the mean. **g** Sensitivities of the Zn/Ag/DNO sensor versus external strains. Inset is the photograph of the sensor being stretched to 100% strain. The scale bar is 8 mm. $n = 3$ for each group. The error bars denote standard deviations of the mean. **h** Dynamic responses of Zn/Ag/DNO sensor to $H_2S$ gas in air or $N_2$ background with reduced concentration from 4 to 0.8 ppm. **i** Sensitivities and theoretical LOD of Zn/Ag/DNO sensor to $H_2S$ in air or $N_2$ background.

conditions by measuring the OCV of the device. After reasonable consideration, we conjecture that the sensing mechanism of our sensor can be attributed to the change of electrode potential under varying $H_2S$ concentrations derived from the reversible chemisorption of gases on the active Ag electrode. In order to intuitively observe the action site of $H_2S$ on the sensor, a selective shielding electrode experiment was carried out by encapsulating the electrode (Ag or Zn) of the sensor and its surrounding gel to isolate the gas from the electrode–hydrogel interface (Fig. 4a). Then, the sensors with encapsulated electrodes were exposed to repeated 2 ppm $H_2S$ and the dynamic responses were compared with that of the unencapsulated Zn/Ag/DNO sensor (Fig. 4b). Obviously, the Zn-encapsulated sensor exhibits a similar dynamic response curve to the unencapsulated sensor, while the response of the Ag-encapsulated sensor is barely observed, even causing a tiny upward fluctuation (about 1 mV) of OCV after encountering $H_2S$. Note that the OCV we measured is expressed as

$$OCV = E_{Ag} - E_{Zn} \qquad (3)$$

where $E_{Ag}$ and $E_{Zn}$ are the electrode potentials of Ag and Zn electrodes, respectively. These results fully demonstrate that the reaction site of $H_2S$ is mainly on the Ag electrode, while only a trace amount of $H_2S$ can be chemically adsorbed on the Zn electrode, which can also lead to a small decrease in $E_{Zn}$ and a small increase in the final OCV.

Generally, there are two processes that may occur at the electrode: the Faradaic process and the non-Faradaic process. In the Faradaic process, the gas undergoes oxidation–reduction reactions at the electrode surface, with the reaction products expected to remain on the electrode surface. In the non-Faradaic process, gas simply adsorbs onto the electrode surface and desorbs from the surface when the gas concentration decreases. On this basis, the interaction of $H_2S$ on the Ag electrode was investigated by semi-in situ X-ray photoelectron spectroscopy (XPS) analysis. We continuously exposed Zn/Ag/DNO sensors to 4 ppm $H_2S$ and extracted the Ag electrodes at different time points for immediate XPS analysis. Two Ag electrode samples were collected, labeled as Sample A and Sample B, representing exposure to $H_2S$ gas for 0 and 0.5 h, respectively (Fig. 4c). The $S\ 2p$ XPS spectra show that there was no residual S element observed on any of the two samples, indicating

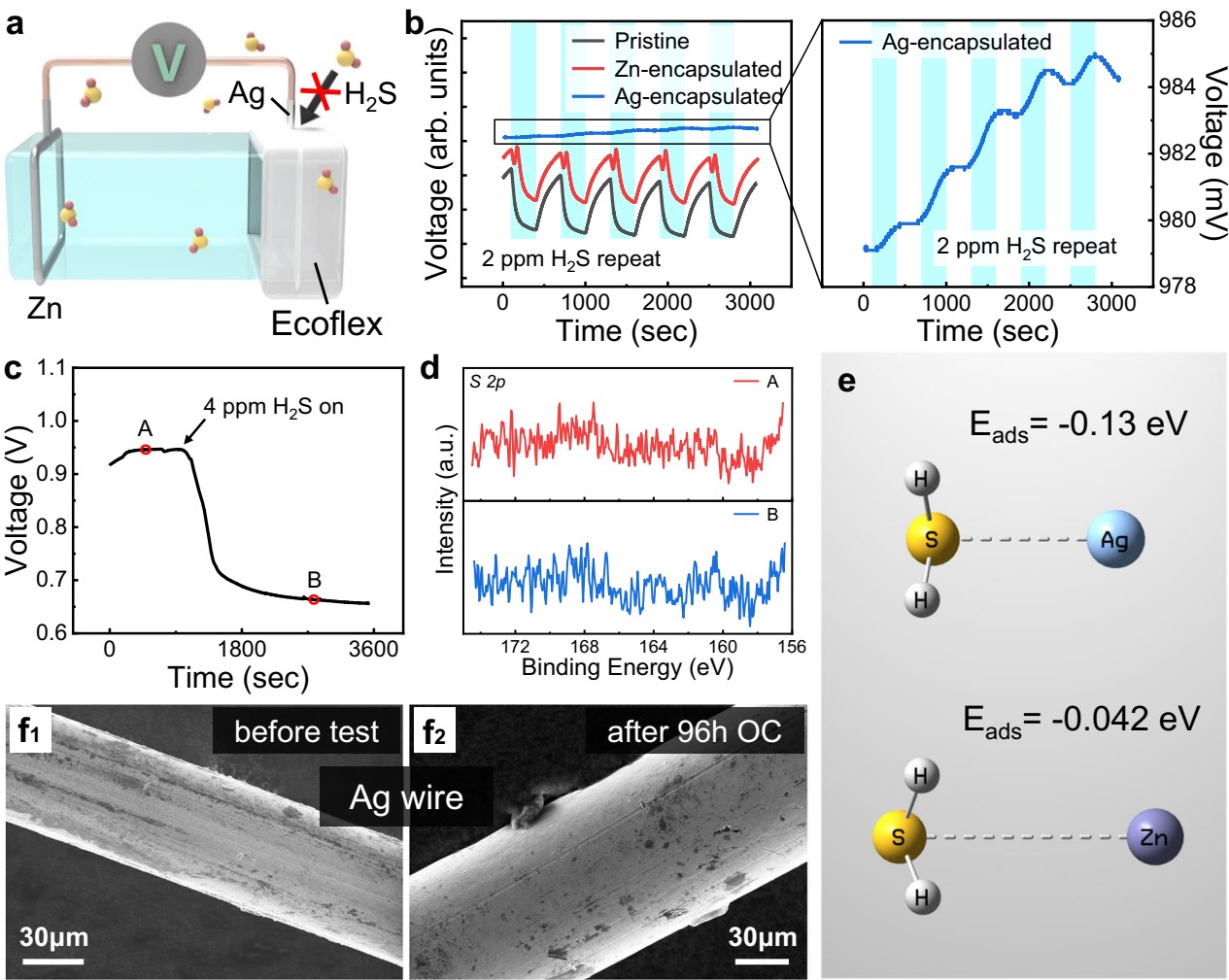

**Fig. 4 | Sensing mechanism of Zn/Ag/DNO. a** Schematic of Zn/Ag/DNO sensor with encapsulated Ag electrode, isolating the H$_2$S molecules from the electrode–hydrogel interface. **b** Dynamic OCV of the Zn/Ag/DNO sensors with encapsulated electrodes to repeated 2 ppm H$_2$S. **c** Continuous exposure of the Zn/Ag/DNO sensors to 4 ppm H$_2$S, labeled with time points A and B, for the XPS analysis of the Ag electrodes. **d** XPS spectrum of Sample A and Sample B, showing S 2p data. **e** H$_2$S adsorbed on the Ag and Zn atoms. The corresponding adsorption energies ($E_{ads}$) were obtained by the DFT calculation. **f** SEM images of the Ag wire (f$_1$) before and (f$_2$) after keeping the Zn/Ag/DNO in OC state for 96 h. The experiment was repeated three times independently with similar results.

that H$_2$S did not undergo oxidation–reduction reactions but rather adsorbed onto the Ag electrode surface and can be easily desorbed during vacuuming (Fig. 4d). Also, XPS analysis of the hydrogel pieces near the Ag and Zn electrodes was then carried out to search for traces of H$_2$S. From the high-resolution S 2p XPS spectrum of the sample close to the Ag electrode, the main peak at 168.5 eV can be ascribed to hexavalent sulfur, referring to the presence of sulfur-containing ions (SO$_4^{2-}$ or SO$_3^{2-}$) (Supplementary Fig. 24a). While for the sample close to the Zn electrode, S element was not observed (Supplementary Fig. 24b). This indicates that H$_2$S adsorbed on the Ag electrode surface can be bonded thereon, and some of them may dissolve into the hydrogel, undergoing ionization reactions. Then the HS$^-$/S$^{2-}$ would be gradually oxidized to the hexavalent state during the subsequent processing steps. On the contrary, H$_2$S is only physically adsorbed on the surface of hydrogel and Zn electrode, and will not cause changes in the valence state of S. The uneven distribution of S elements in the hydrogel demonstrates that the adsorption of H$_2$S occurs at the three-phase interface, rather than through dissolution into the hydrogel and subsequent migration to the electrode–hydrogel interface. After the sensor was exposed to the H$_2$S atmosphere for 10 h, a small amount of S element was found in a particle on the Ag electrode through energy-dispersive spectroscopy (EDS) analysis, while no S elements were detected on Zn electrode and other areas of Ag electrode (Supplementary Fig. 25). The presence of the small particle is believed to be the result of multiple factors over an extended period of time and is not directly related to the sensor's response. Nevertheless, this confirms the characteristic interaction of H$_2$S at the Ag electrode.

Furthermore, theoretical calculations were performed using density functional theory (DFT) to compare the interaction of H$_2$S with Zn and Ag electrodes. As Fig. 4e shows, the adsorption energy between H$_2$S and Ag atoms is calculated to be −0.13 eV, which can be determined as weak chemical adsorption, capable of reversibly causing a change in the electrode potential. Moreover, the negative value indicates that the adsorption of H$_2$S on the Ag electrode is an exothermic process so that the device can still operate even at low temperatures, which is consistent with the previous experimental results. By contrast, the adsorption energy between H$_2$S and Zn atoms is only −0.042 eV, which is mainly physical adsorption, and the weak chemical adsorption that rarely exists can also lead to quite small changes in the Zn electrode potential, thus causing the weak response of the Ag-encapsulated sensor mentioned above[48–51]. To sum up, we propose that the sensing mechanism of the sensor involves the chemical adsorption of H$_2$S at the electrode–hydrogel interface, which

promotes the dissolution of metal ions and leads to a decrease in electrode potential.

It is worth noting that, unlike chemiresistive and amperometric electrochemical sensors, the Zn/Ag/DNO sensor operates with minimal current in the circuit, resulting in minimal electrode degradation and a significantly extended lifespan. To validate this, the Zn/Ag/DNO sensor underwent an extended OCV test (96 h), and the surface morphology of both the Zn and Ag electrodes was examined using scanning electron microscopy (SEM) before and after the test (Supplementary Fig. 26 and Fig. 4f). Notably, no significant corrosion was observed on either electrode, confirming their robustness and stability. The elemental composition of the electrodes was quantitatively analyzed using EDS, and after 96 h, a slight increase in the amount of oxygen (O) was detected on the Zn electrode, which can be attributed to the natural oxidation of Zn in the humid environment (Supplementary Fig. 27). Whereas in the short-circuit (SC) state, only 8 h of continuous testing resulted in severe damage to the Zn electrode due to electrochemical reactions (Supplementary Fig. 28), making it unsuitable for long-term applications. EDS analysis also revealed a significant increase in the amount of O on the Zn electrode, indicating more pronounced corrosion of Zn (Supplementary Fig. 29). To further exclude the occurrence of oxidation–reduction reactions involving $H_2S$ in our sensing system, we stained four Zn/Ag/DNH sensors with a Neutral red-Methylene blue indicator. The sensors were maintained in an SC state or OC state for 7 h, both in air and $H_2S$ (1 ppm) atmospheres (Supplementary Fig. 30). Before the test in the air atmosphere, the hydrogel appeared bluish-purple, showing weak acidity due to the $H^+$ generated by the ionization of carboxyl groups and other groups in the polymer network. During the respective tests, the color near the Ag electrode gradually turned green in the SC state due to the consumption of $H^+$ by electrochemical reduction near the Ag electrode, while the color had basically no change in the OC state. In both states, the color near the Zn electrode gradually turned green, which was attributed to the increase of $OH^-$ in the hydrogel caused by the natural corrosion of Zn in a humid environment. In both SC and OC states in the $H_2S$ atmosphere, the phenomenon of color turning green near the electrodes was observed to weaken. Additionally, there was a slight tendency for the entire surface of the hydrogel to turn slightly bluish-purple, which is attributed to the ionization of $H_2S$ on the gel surface, resulting in the generation of $H^+$. And ionization reactions are distinct from oxidation–reduction reactions.

## Demonstration of Zn/Ag/DNO sensor applied in diverse scenarios

Considering the achieved excellent sensing performance and functionality, as a proof-of-concept, we demonstrated the feasibility of our developed sensor as a portable device for the detection of $H_2S$ biomarkers in several application scenarios, including non-invasive halitosis diagnosis and meat spoilage identification. For halitosis detection, we used gas collection bags to collect sufficient exhalations from a healthy volunteer in two bags, one of which was then mixed with a small amount of $H_2S$ to simulate the breath exhaled by halitosis patients. The concentration of $H_2S$ in simulated halitosis gas was controlled to 400 ppb. Subsequently, exhalations from a halitosis sufferer were collected in the third bag for backup. In the device shown in Fig. 5a, simulated halitosis gas/exhaled gas and dry air were alternately delivered to the Zn/Ag/DNO sensor, and their responses were recorded. To achieve comparable humidity levels between the target gas and background gas, a saturated $K_2SO_4$ solution was employed to humidify the gases. Clearly, the response of the sensor to simulated halitosis gas and healthy exhaled gas was totally different (Supplementary Fig. 31). When the simulated halitosis gas was introduced, the OCV of the sensor dropped by about 9.4 mV due to significant changes in the $H_2S$ content. While as the healthy exhaled gas was introduced, the OCV of the sensor increased by about 4.8 mV, which can be

attributed to the greatly reduced oxygen concentration in exhaled gas compared to air, and the tiny increase in $H_2S$ was not enough to counteract the effect of oxygen. With regard to the exhaled gas from the halitosis sufferer, the OCV of the sensor increased by about 1.06 mV when the gas was introduced. It was evident that the decrease in oxygen concentration caused the OCV of the sensor to rise by about 4.8 mV, and the increase in $H_2S$ concentration caused the OCV to drop by 3.74 mV approximately. Assuming a linear relationship between the response and the concentration of $H_2S$ in the context of human exhaled gas, we calculated the concentration of $H_2S$ in the exhaled breath of the halitosis sufferer to be 105 ppb, which is a reasonable value relative to clinical data (Fig. 5b). As a result, this demonstration experiment convincingly shows the viability of the Zn/Ag/DNO sensor for timely and non-invasive halitosis diagnosis. Besides, the sensors could be also used to monitor the freshness of meat. Specifically, in the experimental group, we put the Zn/Ag/DNO sensor in a closed gas bottle containing a piece of fresh pork and stored it in a −18 °C refrigerator at first and then brought it to RT (22 °C) after 121.7 h, and the OCV of the device was measured at intervals. For comparison, another sensor fabricated in the same batch was placed in an empty closed gas bottle as the blank group (Fig. 5c). As shown in Fig. 5d, the OCV of the sensor was stable at 0.99 V when stored in the refrigerator, demonstrating that the pork was in a fresh state. Once the gas bottle was removed from the refrigerator, the OCV began to drop, indicating that the pork was gradually spoiling and releasing $H_2S$. Then at 246.5 h, we removed the spoiled pork from the gas cylinder, exposing the sensor to the surrounding environment. The OCV rose rapidly from 613 to 750 mV due to the dilution of $H_2S$. For the blank group, the OCV remained around 0.86 V throughout the process, which indicates that the variation of the OCV of the device in the experimental group is completely caused by meat spoilage, confirming its broad application prospect in the food industry.

Furthermore, a wireless $H_2S$ sensing system was designed and developed in combination with intelligent technology, which consisted of the developed sensor and a circuit module for data processing and transmission (Supplementary Fig. 32). First of all, with the advantage of self-powering, the sensor can continue to collect signals spontaneously, with low power consumption. Then, the signal processed by the circuit module can be transmitted to and displayed on a mobile phone or a computer through Bluetooth to realize wireless, timely, and convenient observation. Also, the transmitted data can be uploaded to the cloud synchronously, thus allowing remote $H_2S$ monitoring through multiple terminals (Fig. 5e). Users can download real-time data from the cloud whenever they are in a moving car, in an apartment, or during exercise through an App that we programmed. Ultimately, the assembled whole system was only the size of a coin, which was very beneficial to the development of portable products. Multi-terminal remote real-time monitoring of whether $H_2S$ leakage occurs in places of interest such as laboratories and factories could be realized. For demonstration, we employed this system to identify whether $H_2S$ leakage occurs in a closed environment and give an alarm in real-time. Wherein, the dynamic response curve could be displayed on the mobile phone's self-programmed App through Bluetooth transmission, and a threshold of 650 mV was set in advance. Before exposure to $H_2S$, the recorded voltage stayed at above 720 mV and the phone showed NORMAL in green. When the $H_2S$ was on, the voltage began to drop. And the phone displayed a red ALARM sign once the voltage fell below the threshold. When the $H_2S$ was off, the voltage rose and returned to a level above the threshold, leading to the reappearance of the green NORMAL sign (Supplementary Movie 2 and Fig. 5f). Furthermore, with the help of the cloud, the $H_2S$ concentration in the lab can be monitored in real-time by users far away in the office with a tablet, so authorized users can be alerted anywhere with an internet connection in the event of $H_2S$ leakages in the laboratories (Supplementary Movie 3 and Fig. 5g). Not only that, thanks to the excellent

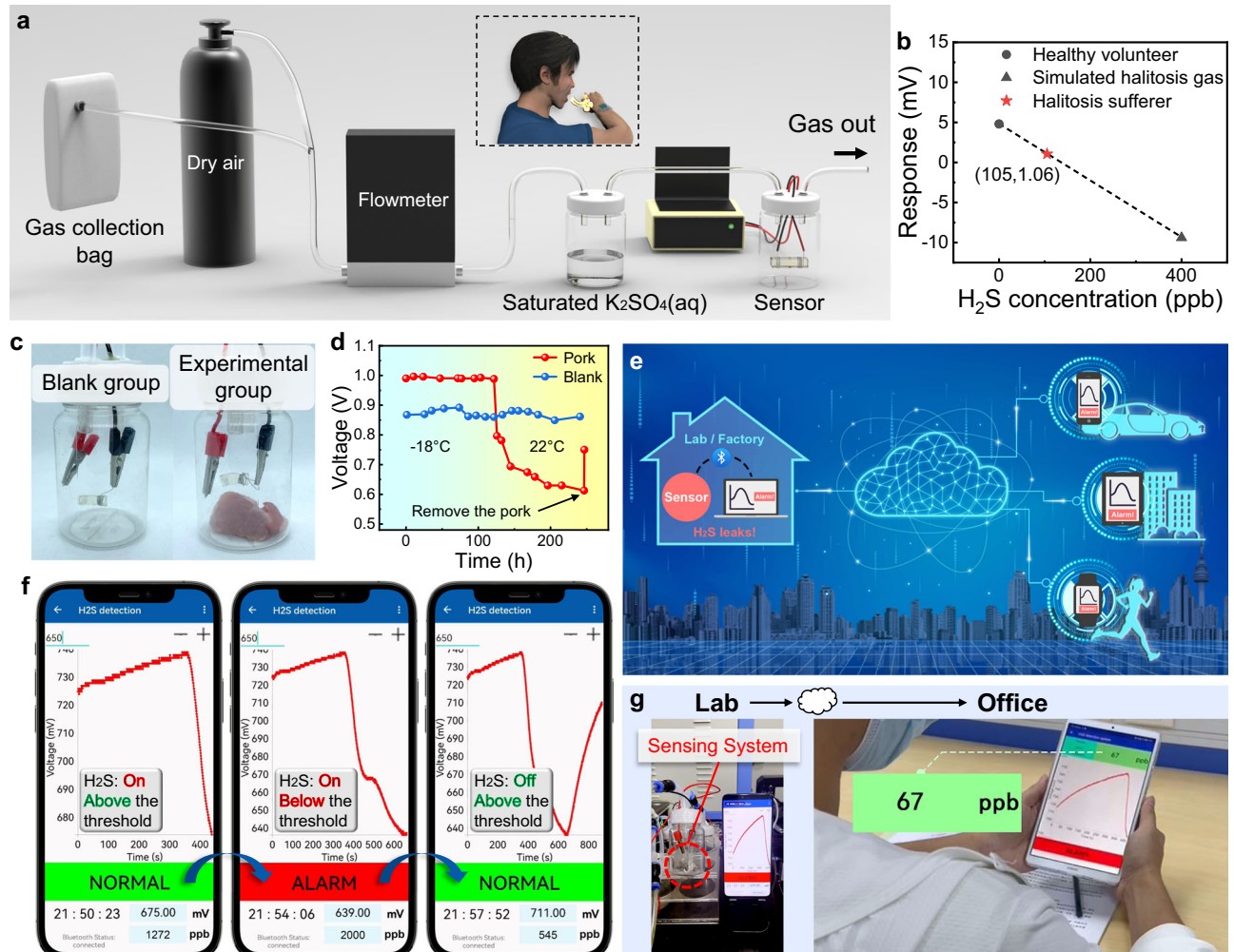

**Fig. 5 | H₂S detection application scenarios. a** Schematic diagram of halitosis diagnosis experimental device. **b** Response versus H₂S concentration curve of the Zn/Ag/DNO sensor to the exhaled breath of a healthy volunteer/halitosis sufferer and simulated halitosis gas. **c** Photographs of the blank group and experimental group in pork spoilage monitoring. **d** Bottle with a piece of pork and blank groups were moved from the refrigerator (−18 °C) to RT (22 °C). The OCV of the sensors was recorded at intervals. **e** Schematic diagram of remote monitoring of H₂S concentration in laboratories/factories using cloud technologies. **f** Wireless alarm demonstration by an App in a smartphone using Bluetooth technology. **g** Remote alarm demonstration: user was in the office while monitoring the H₂S levels in the lab.

sensing performance and environmental adaptability of our sensor, this designed wireless H₂S monitoring system is also expected to be applied in more application scenarios and fields, such as food transportation, mine exploration, portable medical equipment, etc., and it will play an important role in the development of the Internet of Things and CMT.

## Discussion

In summary, a flexible, room-temperature operable, and self-powered gas sensor was proposed based on the susceptibility of the electrode potential to the target gas, with the advantages of good sensing performance and wide environmental adaptability. Given its unique attributes, this sensor could serve as an ideal candidate for wearable electronics. Comprised of two metal electrodes and a solid hydrogel electrolyte derived from a galvanic cell, the sensor exhibits a straightforward and cost-effective design, enabling arbitrary stretching and bending due to the intrinsic flexibility of the electrolyte. During testing, the OCV of the device is measured without an external power supply, and the value is the difference in the electrode potential between two electrodes at the hydrogel surface. When exposed to the target gas, the OCV of the sensor changes accordingly, and its

response mechanism is attributed to the reversible weak chemisorption of gas on the electrode surfaces. By utilizing the Ag wire as the active electrode, the resulting sensor exhibits excellent sensing performance for H₂S, including high sensitivity (23.7 mV/ppm), low LOD (0.79 ppb), excellent selectivity, and good repeatability and stability. Also, the sensor demonstrates an immunity to disturbances and remains functional at sub-zero temperatures, in anaerobic environments, and under tensile deformation, which is highly competitive with existing H₂S sensors. Through a series of experiments and DFT calculations, the specific reversible interaction of H₂S on the Ag electrode is confirmed, leading to the responsiveness of the device. Thanks to the excellent sensing performance achieved by this device, it is then used for halitosis diagnosis and identification of meat spoilage, fully capable of convenient and timely biomarker detection. Furthermore, a compact wireless sensing system is designed and developed, capable of wirelessly transmitting the detected signals to the user terminal through Bluetooth or cloud-sharing technology, enabling real-time, and remote H₂S monitoring. This work presents a pioneering concept and fundamental basis for the future development of low-cost, self-powered, and wearable gas sensors with high performance at RT. This significant advancement holds great promise in enhancing human

health and safety, contributing to continuous progress in the field of sensor technology.

## Methods

### Synthesis of PAM/CA DN hydrogels and organohydrogels

The PAM/CA DN hydrogel was synthesized in two steps (Fig. 1a)[52]. Firstly, the reactants, including acrylamide (AM), sodium alginate (SA), N,N'-methylene bisacrylamide (MBA), and ammonium persulphate (APS) were successively added into deionized water with the weight ratios of 8:1:0.005:0.05. The mixture with a total weight of 50 g was stirred at 550 rpm for 30 min to attain complete dissolution. To accelerate the gelling process, 20 μL N, N, N', N'-tetra-methylethylenediamine was added into the solution to function as an accelerator. Then the solution was poured into a petri dish and heated at 65 °C for 2 h to form the polyacrylamide network. Secondly, the semi-finished hydrogel was cut into pieces with a size of $15 \times 6 \times 6$ mm, which were subsequently immersed in 1 mol/L $CaCl_2$ solution for 3 h at 25 °C. $Ca^{2+}$ would diffuse into the hydrogel network under the concentration gradient, leading to the crosslinking of SA molecules centered on the introduced $Ca^{2+}$. Here, the DN hydrogel consisting of PAM and CA networks was successfully fabricated. To obtain the organohydrogels, the as-synthesized DN hydrogel pieces were immersed into 10 mL Gly for a period, during which Gly diffused into the DN hydrogel to form a Gly–water binary solvent.

### Synthesis of Zn/Ag/DNH and Zn/Ag/DNO sensor

A Zn/Ag/DNH sensor was prepared by winding Zn wire and Ag wire at the two ends of a DN hydrogel block. Unless otherwise specified, the Zn wire has a diameter of 0.3 mm and the Ag wire has a diameter of 0.06 mm, with a winding count of three turns for both. By replacing the DN hydrogel in the Zn/Ag/DNH sensor with a DN organohydrogel, a Zn/Ag/DNO sensor can be obtained.

### Synthesis of hydrogel samples for XPS analysis

A Zn/Ag/DNH sensor was exposed to 2 ppm $H_2S$ for 10 h. Subsequently, the hydrogel from the sensor was extracted, and hydrogel blocks were cut from the regions where they were wrapped around the Zn and Ag electrodes. The hydrogel blocks were then placed in a drying oven at 110 °C for 2 h. The obtained dried hydrogel blocks were subjected to XPS analysis.

### Materials characterization

The transmittance spectra were obtained using a UV–visible spectrophotometer (Thermo Fisher, Inc., Evolution 220) with a wavelength of 110–1190 nm. The strain–stress curves were acquired on a motorized stretching stage, which was driven by a Zolix SC300-3A motion controller. The FTIR spectra were acquired by Fourier transform infrared spectroscopy-microscope (Thermo Fisher, NICOLET 6700) with the wavenumber of 400–4000 cm⁻¹. The mass of samples was acquired on an electronic scale (Mettler Toledo, MS204S). The DSC spectra were acquired using a differential scanning calorimeter (Netzsch, DSC-204) at a cooling rate of 5 °C/min from 15 to −120 °C with nitrogen flow. The SEM images and EDS results were obtained using a field emission scanning electron microscope (Wavetest, SUPRA 60). The XPS spectra were obtained on an XPS machine (Thermo Fisher, Escalab 250).

### Gas sensing measurement

The sensors were placed in a sealed chamber for measuring the gas sensing properties at RT (25 °C) unless otherwise noted. An electrochemical workstation (CH Instruments, CHI760E) was used to record the OCV of the gas sensors. Different concentrations of test gases such as $H_2S$, $O_2$, NO, $NH_3$, and $CO_2$ were obtained by diluting the 10 ppm $H_2S$, 200,000 ppm $O_2$, 9.5 ppm NO, 50 ppm $NH_3$, and 1000 ppm $CO_2$, respectively, with $N_2$ or air. The background, diluting, and purging gases were kept consistent ($N_2$ or air) during each sensing event. In the figures of this paper, the cyan-shaded part represents the introduction of the target gas, and the other parts represent the introduction of the background gas. The flow rate was controlled by the digital mass flow controllers (MFC) with a total rate of 250 sccm unless otherwise noted. The high concentrations of VOCs (e.g., 1000 ppm ethanol, 1000 ppm isopropanol, 1000 ppm acetone) were obtained using a DGL-III gas and liquid distribution system (China ELITE TECH). The humidification of gas was achieved via a bubbling method[53]. Data were analyzed through Origin 2022b edu.

### Demonstration of $H_2S$ sensor on halitosis detection

Gas collection bags were used to collect sufficient exhalations from a healthy volunteer and a halitosis sufferer, respectively. The collected gases and dry air were alternately delivered to the Zn/Ag/DNO sensor, which was set in a gas bottle. The OCV of the sensor was recorded by an electrochemical workstation (CH Instruments, CHI760E). A healthy subject and a halitosis sufferer, both male, provided the exhalations used in the halitosis diagnostic test. The two subjects' ages ranged from 24 to 25.

### Demonstration of $H_2S$ sensor on meat freshness monitoring

Two Zn/Ag/DNO sensors were prepared, one in a closed gas bottle with a piece of fresh pork as the experimental group, and the other in an empty closed gas bottle as the blank group. Both groups were stored in a −18 °C refrigerator for 121.7 h and then transferred to RT (22 °C) for 124.8 h. Finally, the spoiled pork was removed from the gas bottle. The OCV of the devices was measured by an electrochemical workstation at intervals.

### Ethics declarations

The data were obtained with the informed consent of all participants. This study was approved by the Institutional Review Board of Sun Yat-sen University and was conducted in accordance with the Declaration of Helsinki (KQEC-2023-32-02).

### Reporting summary

Further information on research design is available in the Nature Portfolio Reporting Summary linked to this article.

## Data availability

The data generated in this study have been deposited in the Science Data Banke database under the accession code https://www.scidb.cn/s/IbiIza.

## Code availability

The source codes for OCV signal recording and Bluetooth transmission used in this study are available at https://github.com/593773740/H2S-sensing-device.

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

## Acknowledgements
J.W. acknowledges financial support from the National Natural Science Foundation of China (61801525), Guangdong Basic and Applied Basic Research Foundation (2020A151501069), and Fundamental Research Funds for the Central Universities, Sun Yat-sen University (22lgqb17).

## Author contributions
J.W. supervised and directed the research. J.W. and W.X.H. conceived the idea and designed the research. W.X.H. carried out and participated in all the experiments and wrote the initial manuscript. Q.L.D. participated in the Mechanical properties measurement of hydrogels. W.X.H. and J.W. performed data analyses. W.X.H., Q.L.D., and J.W. revised the manuscript. H.W. provided the BLE module and designed the App. Z.X.W. and Y.B.L. participated in the analysis of the results. W.X.S. contributed to the theoretical calculations. Y.J.L., L.Y., and C.L. contributed to the design and discussion of the manuscript and the results. All authors discussed the results.

## Competing interests
The authors declare no competing interests.
