## [Peer Review File · Nature Communications]

Design of stretchable and self-powered sensing device for portable and remote trace biomarkers detectionReviewers' Comments:

Reviewer #1:

Remarks to the Author:

In this manuscript, the H₂S sensing system is constructed based on organohydrogels and metal electrode. The work is complete, demonstration is substantial, and the discussion is detailed. But the original innovation is slightly deficient and needs to be further improved. The structure of gel-like electrolyte and metal electrodes in flexible battery-type gas sensors has been reported in other papers, for example, *ACS Appl. Mater. Interfaces* 2021, 13, 39, 46507- 46517, and the gel modification schemes described in this paper are derived from literature reports. In addition, there are some issues that need to be discussed.

1. The author mentioned in the original article that 'After H₂S was removed, the OCV of the device gradually recovered, indicating the weak reversible chemisorption of H₂S on the electrode rather than chemical reactions', which requires more intuitive experimental evidence. First, the adsorption of H₂S on metal electrodes at room temperature should be verified by experimental results (EDS results in Fig. 4c cannot be used to illustrate the adsorption of H₂S on metal electrodes at room temperature. Fig. 4c is the result of long-term action with multiple possible influencing factors and cannot represent the transient sensitive response mechanism). Why the metal electrode can specifically adsorb H₂S, but not the other gases in the selectivity test? Secondly, the authors use the comparison of Neutral red-Methylene blue indicators in SC state and OC state to illustrate that there is no electrochemical reaction. What is the concentration of H₂S during the test? In addition, the authors should supplement the blank sample test in the SC state to verify that the consumption of H⁺ at Ag electrode side is caused by the electrochemical reaction of H₂S.
2. In the application demonstration, the author designed a variety of scenarios, but the gas types in the selectivity test were not comprehensive.
3. How exactly is the self-powered feature implemented? The text does not explain in detail.
4. Line 98 of the main text, expressions are no longer appropriate. The authors write "Nevertheless, research on hydrogel-based gas sensors is relatively infrequent, and hydrogel-based H₂S sensors have not even been invented so far." In fact, Wang et al. invented the Cu-PAN hydrogel H₂S sensor. (*Sensors & Actuators: B. Chemical* 376 (2023) 132968; <https://doi.org/10.1016/j.snb.2022.132968>)
5. The response value of the sensor was greatly influenced by the change of humidity and temperature, how to remove all the distractions?
6. A rough estimate from the response recovery curves of the sensors shows that the response/recovery times are all slow, but the authors do not mention this. Furthermore, the theoretical LOD of 0.73 ppb H₂S for the gas sensor is not obtained from the specific test curves, which can't be convincing enough.
7. Only one measurement of the sensing characteristics of the sensor at different humidities, temperatures and strain were reported for each concentration, please provide mean value and standard deviation in order to evaluate sensor repeatability.

Reviewer #2:

Remarks to the Author:

In this manuscript, the authors reported a flexible and self-powered sensing device based on the structure of galvanic cells for monitoring of H₂S. The OCV is measured without an external power supply. Moreover, a water-rich and ion-conducting PAM/CA hydrogel was introduced as the solid-state electrolyte for improving the flexibility and stretchability of the system. Overall, the method demonstrated shows a decent prospect of real-time and remote monitoring of H₂S. I think this work will attract readers of *Nature Communications*, below are some detailed concerns before the manuscript can be published. Thus, I recommend this work to be published on *Nature Communications* after major revision.

It is pointed out that the sensor is self-powered, and there should be more corresponding descriptions and a physical picture should be added.

SEM diagrams of the DNH should be given to show its microstructure.

The authors chose PAM and CA to synthesize DNH. How about some other skeleton materials?

Considering the mechanical properties and optical transparency, there should be many other candidates.

In Supplementary Fig. S1, it would be more convincing that using videos instead of photographs to show how the hydrogel changes in tensile strains.

For comparison, the long-term test of pristine hydrogel should be added in Supplementary Figure S12.

The authors stated that the developed sensor has excellent selectivity, but failed to give the reason. A detailed explanation is highly suggested, e.g. the adsorption energies between different gases and electrode metal based on DFT calculation.

Reproducibility is the key parameter in evaluating sensor performance, and therefore it should be explored as well.

H₂S in several application scenarios, including halitosis diagnosis and meat spoilage identification. The comparison between the data obtained by the sensor and the laboratory equipment should be given.

The error bars should be added such as in Fig. 2D and Supplementary Figure S4.

It should be "*V*_(H_2 S)" instead of "*V*_(H_2 S)". Please note that all the variables should be displayed in italic form.

It is suggested that the English expression can be further polished by a language expert.

Several recently published work on room-temperature gas sensors are recommended for citation.

(e.g. Chemical Engineering Journal, 2022, 446: 136937; ACS Sensors, 2022, 7 (4): 1183-1193;

Analytical Chemistry, 2020, 92 (16): 11277-11287).

Reviewer #3:

Remarks to the Author:

In this manuscript, Wu and coworkers report a flexible and self-powered H₂S gas sensing device by adopting the structure of galvanic cells and using stretchable organohydrogels as solid-state electrolytes. The device shows superior gas sensing performance at room temperature and maintains its functionality at sub-zero temperatures or under mechanical deformations, which is unattainable by existing H₂S sensors. The demonstrations of halitosis diagnosis, meat freshness monitoring and real-time remote H₂S leakage alarm reveal huge application potential in human health and safety protection. Additionally, this paper provides a novel strategy for the design of flexible, self-powered gas sensors and creates the necessary groundwork for subsequent research. In general, this work is very interesting and worthy of investigation, therefore I would like to recommend the publication in Nature Communications after addressing the following minor issues.

1. According to Supplementary Fig. S18 and Fig. 2i, both H₂S and O₂ decrease the open circuit voltage of the sensor. However, H₂S is a reducing gas while O₂ is an oxidizing gas. Is the adsorption site and effect of O₂ molecules on the sensor the same as H₂S? The effect of oxygen adsorption on the electrode potential deserves further investigation.

2. The EDS spectrum is missing in Supplementary Fig. S23-24 and Fig. 4f. The author should add the EDS spectrum to quantitatively analyze the electrode composition changes before and after the experiment.

3. How to guarantee the uniformity of the developed device, which is very important for the commercialization of the device. In addition, the newly proposed sensing device is millimeter-sized, and whether it can be further miniaturized?

4. Normally, water-containing hydrogels cannot be used directly for XPS analysis because the vacuum of the cavity is not up to the test requirements. What treatment was given to the hydrogel samples in the article (Fig. 4c-d)? It is recommended that the preparation process of the hydrogel samples for XPS analysis be supplemented in the "Methods" section.

5. The result in Fig. 3b shows that the response of the sensor increases with the increase of relative humidity. While in the halitosis detection experiment (Fig. 5a), the humidity of dry air and the

collected exhaled gas is significantly different. Does the difference in humidity between the two gases affect the experimental results? Please explain this in detail.

Responses to the reviewers' comments

Manuscript ID: NCOMMS-23-02667A

Title: Design of novel stretchable and self-powered sensing device for portable and remote trace biomarkers detection

Dear Reviewers:

Thank you for reviewing our manuscript and giving us valuable comments and feedback so that the quality of our manuscript can be improved. Based on the reviewers' comments, we have executed supplementary experiments and revised our manuscript carefully. Our responses are given and corresponding revisions are highlighted in the revised manuscript.

Point-by-point responses to the Reviewers:

Reviewer #1 (Remarks to the Author):

In this manuscript, the H₂S sensing system is constructed based on organohydrogels and metal electrode. The work is complete, demonstration is substantial, and the discussion is detailed. But the original innovation is slightly deficient and needs to be further improved. The structure of gel-like electrolyte and metal electrodes in flexible battery-type gas sensors has been reported in other papers, for example, ACS Appl. Mater. Interfaces 2021, 13, 39, 46507- 46517, and the gel modification schemes described in this paper are derived from literature reports. In addition, there are some

issues that need to be discussed.

Response:

Thanks for the valuable comments from reviewer. Although our sensor shares similarities in structure with the article mentioned by the reviewer, it is fundamentally different in terms of the detected gas, sensing performance, and underlying mechanism. First of all, our target gas is H₂S, a toxic gas generated from crude oil, natural gas, bacterial decomposition of organic matter, and industrial activities. Meanwhile, it is a biomarker for halitosis and liver diseases. Portable and real-time accurate detection of low concentrations of H₂S gas holds significant implications for hazard warning, disease prevention, and food safety. Previously, there have been no reports on the use of hydrogel for electrical H₂S sensors, and we have achieved this for the first time. Furthermore, our sensor exhibits a lower detection limit compared to most existing H₂S gas sensors. Operating at room temperature, being self-powered, flexible, and transparent, our sensor is exceptionally well-suited for the advancement and implementation of wearable electronics. In contrast, the Zn-air battery device in "ACS Appl. Mater. Interfaces 2021, 13, 39, 46507- 46517" is a conventional electrochemical sensor based on gas electrochemical reactions, and requires an external power source to supply current during operation. While the response of our sensor to H₂S is based on the gas's chemical adsorption, and the open-circuit voltage (OCV) serves as the output signal, solely dependent on the electrode potential without the need for external power supply. Most importantly, we have demonstrated the sensor's applicability in halitosis detection and meat quality monitoring. By utilizing Bluetooth and cloud technologies,

we have further expanded its application scope, revealing the sensor's immense potential in cloud healthcare, food safety, and industrial safety.

1. The author mentioned in the original article that ‘After H₂S was removed, the OCV of the device gradually recovered, indicating the weak reversible chemisorption of H₂S on the electrode rather than chemical reactions’, which requires more intuitive experimental evidence. First, the adsorption of H₂S on metal electrodes at room temperature should be verified by experimental results (EDS results in Fig. 4c cannot be used to illustrate the adsorption of H₂S on metal electrodes at room temperature. Fig. 4c is the result of long-term action with multiple possible influencing factors and cannot represent the transient sensitive response mechanism). Why the metal electrode can specifically adsorb H₂S, but not the other gases in the selectivity test? Secondly, the authors use the comparison of Neutral red-Methylene blue indicators in SC state and OC state to illustrate that there is no electrochemical reaction. What is the concentration of H₂S during the test? In addition, the authors should supplement the blank sample test in the SC state to verify that the consumption of H⁺ at Ag electrode side is caused by the electrochemical reaction of H₂S.

Response:

Thanks for the valuable comments from reviewer. Next, we will discuss each of these issues one by one.

Firstly, there are generally two processes that may occur at the electrode: the Faradaic process and the non-Faradaic process. In the Faradaic process, gas undergoes oxidation-

reduction reactions at the electrode surface, with the reaction products expected to remain on the electrode surface. In the non-Faradaic process, gas simply adsorbs onto the electrode surface and desorbs from the surface when the gas concentration decreases. The selective shielding electrode experiment in Figure 4a-b has demonstrated that the active sites for H₂S are mainly on the Ag electrode. To investigate the transient sensitive response mechanism of the sensor, we continuously exposed Zn/Ag/DNO sensors to 4 ppm H₂S and performed semi-in situ XPS analysis on different time-point Ag electrodes. Two Ag electrode samples were collected, labeled as Sample A and Sample B, as shown in Figure 4c. The S 2p XPS spectra of the two samples showed no presence of S element, indicating that H₂S did not undergo oxidation-reduction reactions but rather adsorbed onto the Ag electrode surface. The presence of S element observed in Supplementary Figure S24 and Supplementary Figure S25 is the result of long-term effects from multiple possible influencing factors.

Corresponding discussion has been supplemented in the revised manuscript. As shown in page 20-21: “Generally, there are two processes that may occur at the electrode: the Faradaic process and the non-Faradaic process. In the Faradaic process, gas undergoes oxidation-reduction reactions at the electrode surface, with the reaction products expected to remain on the electrode surface. In the non-Faradaic process, gas simply adsorbs onto the electrode surface and desorbs from the surface when the gas concentration decreases. On this basis, the interaction of H₂S on the Ag electrode was investigated by semi-in situ X-ray photoelectron spectroscopy (XPS) analysis. We continuously exposed Zn/Ag/DNO sensors to 4 ppm H₂S and extracted the Ag

electrodes at different time points for immediate XPS analysis. Two Ag electrode samples were collected, labeled as Sample A and Sample B, representing exposure to H₂S gas for 0 h and 0.5 h, respectively (Fig. 4c). The S 2p XPS spectra shows that there was no residual S element observed on any of the two samples, indicating that H₂S did not undergo oxidation-reduction reactions but rather adsorbed onto the Ag electrode surface and can be easily desorbed during vacuuming (Fig. 4d).”

Secondly, the metal electrode specifically adsorbed H₂S but not other gases because of the inherent properties of Ag and Zn. In fact, the electrode potential of the metal depends on the dissolution of metal ions in the electrolyte, and a detailed discussion on this matter is provided in the revised manuscript on page 11. Thus, the response of the sensor relies on the gas's influence on the dissolution of metal ions. Taking H₂S as an example, the metal electrode adsorbs H₂S, which promotes the dissolution of metal ions on the electrode. The dissolved metal ions form a double layer at the electrode-hydrogel interface, causing a decrease in electrode potential due to the loss of metal cations. Other gases have a minor influence on the dissolution of metal ions on the electrode, resulting in either a small response or no response from the sensor. This can be verified by examining the electrochemical impedance spectroscopy (EIS) of the sensor under different gas atmospheres (Supplementary Fig. S22). We conducted EIS tests on the sensor in pure N₂, 4 ppm H₂S, 40000 ppm O₂, 4 ppm NO, and 1000 ppm ethanol, respectively, which could reflect the constant phase angle element (CPE) and double-layer capacitance at the electrode-hydrogel interface. After fitting the data, the CPE-T value for the sensor in the presence of H₂S was determined to be 33.8 μF, higher than

that under other gas atmospheres. This indicates that the double layer under an H₂S atmosphere accumulates more charge, which originates from the dissolution of metal ions on the electrode.

Corresponding discussion has been supplemented in the revised manuscript. As shown in page 16: “And this could be attributed to the sulfurophilic nature of Ag and the promoting effect of H₂S on the dissolution of metal ions from the Ag electrode, which was further validated by the electrochemical impedance spectroscopy (EIS) of the sensor under different gas atmospheres (Supplementary Fig. S22).”

Thirdly, in the original manuscript, the testing was conducted in air, which corresponds to the blank group mentioned by the reviewer. The consumption of H⁺ is caused by the reaction of the galvanic battery (reaction 1 and 2) rather than the electrochemical reaction of H₂S.

In the OC state, it is difficult for electrons to transfer from the Zn electrode to the Ag electrode through the external circuit, making reactions 1 and 2 less likely to occur. As a result, Zn can only slowly undergo a corrosion reaction with the surrounding H⁺, depleting the surrounding H⁺ concentration. This explains why the hydrogel around the Ag electrode turns green in the SC state, while the hydrogel around the Zn electrode turns green in the OC state. Additionally, the galvanic battery reaction enhances the reaction rate, causing the hydrogel around the Ag electrode to exhibit a more pronounced green color in the SC state compared to the hydrogel around the Zn

electrode in the OC state.

The original manuscript might have some ambiguities, so we have made revisions and supplemented the testing in a 1ppm H₂S atmosphere. Corresponding discussion has been supplemented in the revised manuscript. As shown in page 23: “To further exclude the occurrence of oxidation-reduction reactions involving H₂S in our sensing system, we stained four Zn/Ag/DNH sensors with a Neutral red-Methylene blue indicator. The sensors were maintained in a SC state or OC state for 7 h, both in air and H₂S (1 ppm) atmospheres (Supplementary Fig. S30). Before the test in air atmosphere, the hydrogel appeared bluish-purple, showing weak acidity due to the H⁺ generated by the ionization of carboxyl groups and other groups in the polymer network. During the respective tests, the color near the Ag electrode gradually turned green in the SC state due to the consumption of H⁺ by electrochemical reduction near the Ag electrode, while the color had basically no change in the OC state. In both states, the color near the Zn electrode gradually turned green, which was attributed to the increase of OH⁻ in the hydrogel caused by the natural corrosion of Zn in a humid environment. In both SC and OC states in H₂S atmosphere, the phenomenon of color turning green near the electrodes was observed to weaken. Additionally, there was a slight tendency for the entire surface of the hydrogel to turn slightly bluish-purple, which is attributed to the ionization of H₂S on the gel surface, resulting in the generation of H⁺. And ionization reactions are distinct from oxidation-reduction reactions.”

Fig. 4 **c** Continuous exposure of the Zn/Ag/DNO sensors to 4 ppm H₂S, labeled with time points A and B, for the XPS analysis of the Ag electrodes. **d** XPS spectrum of Sample A and Sample B, showing S 2p data.

Supplementary Figure S22. Electrochemical impedance spectroscopy (EIS) analysis of the Zn/Ag/DNO sensor under different gas atmospheres. a Impedance spectra; **b** CPE-T values obtained by fitting. The insert is an equivalent circuit diagram of the Zn/Ag/DNO sensor.

Supplementary Figure S30. PH changes of hydrogels near Zn (left) and Ag (right) electrodes explored by Neutral red-Methylene blue indicator. a The color of Neutral red - Methylene blue indicator corresponding to PH. **b-c** Photographs showing the color evolutions of Zn/Ag/DNHs after being stained with Neutral red-Methylene blue indicator and kept in SC state or OC state, in air (b) and H₂S (1 ppm) atmospheres (c).

2. In the application demonstration, the author designed a variety of scenarios, but the gas types in the selectivity test were not comprehensive.

Response:

Thanks for the valuable comments from reviewer. According to the reference, human

breath contains 694 compounds in respiratory analysis¹. Apart from H₂S, the exhaled breath of individuals with halitosis may also contain gases such as acetone, isoprene, methanethiol, and dimethyl sulfide. During the process of pork decomposition, bacteria break down proteins, resulting in the production of gases such as H₂S, methanethiol, dimethyl sulfide, NH₃, methane, and CO₂. In our previous manuscript, we tested gases including H₂S, NH₃, CO₂, and acetone. Now we have supplemented tests for methanol, isoprene, trichloromethane, and dimethyl sulfide, aiming to make the gas types covered in the selective testing as comprehensive as possible. As shown in Supplementary Fig. S21, the sensor exhibited a response of -10.69 mV to dimethyl sulfide, while showing no response to methanol, isoprene, and trichloromethane, demonstrating excellent selectivity as before. Among the main interfering gases in application scenarios, methanethiol and methane are still lacking due to limited experimental conditions, and it is expected to be supplemented in further research.

We have supplemented the corresponding discussion in the revised manuscript. As shown in page 16: “In this case, we investigated the response of the Zn/Ag/DNO sensor to some possible interfering gases in the environment or in exhaled air at 25 °C, 58% RH, including 40000 ppm O₂, 1.9 ppm NO, 200 ppm CO₂, 10 ppm NH₃, 1000 ppm ethanol, 1000 ppm isopropanol, 1000 ppm acetone, 1000 ppm methanol, 1000 ppm isoprene, 1000 ppm trichloromethane, and 1000 ppm dimethyl sulfide (Supplementary Fig. S21). Although 40000 ppm O₂ reduced the OCV of the sensor by 12.47 mV, it had little effect on the detection of H₂S when the O₂ concentration changed little. Furthermore, the sensor demonstrated a response of -10.69 mV to 1000 ppm dimethyl

sulfide, implying its preference for sulfur-containing gases. Except for O₂ and dimethyl sulfide, the gas sensor exhibited a negligible response of 0.13 mV to 1.9 ppm NO and imperceptible responses to other interfering chemicals, demonstrating the excellent selectivity (Fig. 2i).”

Supplementary Figure S21. Selectivity of Zn/Ag/DNO sensor. **a** Plot showing the process that controlled the periodic “on” and “off” of target gases to Zn/Ag/DNO sensor in (b-l). **b-l** OCV of the Zn/Ag/DNO sensor in the cyclic detection of **b** 40000 ppm O₂, **c** 1.9 ppm NO, **d** 200 ppm CO₂, **e** 10 ppm NH₃, **f** 1000 ppm ethanol, **g** 1000 ppm isopropanol, **h** 1000 ppm acetone, **i** 1000 ppm methanol, **j** 1000 ppm isoprene, **k** 1000 ppm trichloromethane, and **l** 1000 ppm dimethyl sulfide.

1. Van den Velde, S., van Steenberghe, D., Van Hee, P., Quiryne, M. Detection of odorous compounds in breath. *J. Dent. Res.* **88**, 285-289 (2009).

3. How exactly is the self-powered feature implemented? The text does not explain in

detail.

Response:

Thanks for the valuable comments from reviewer. Metal lattices are known to contain ordered metal ions and mobile electrons. When a metal electrode is immersed in water, the highly polar water molecules interact with the metal ions in the lattice, causing a process known as hydration. This process weakens the bonding between some of the metal ions and other metal ions present in the metal, leading to the departure of some metal ions from the metal surface and into the surrounding water layer, known as ion dissolution. The metal electrode loses metal ions and becomes negatively charged, while the solution becomes positively charged due to the entry of metal ions. The opposing charges attract each other, causing most of the metal ions to accumulate in the water layer near the metal electrode. However, the repulsion of metal ions near the water layer and the adsorption of electrons on the electrode hinder the continued dissolution of the metal until an equilibrium state is reached.

When a metal electrode is immersed in a solution containing a metal salt, if the chemical potential of the metal ions in the solution phase is higher than that in the electrode phase, the metal ions will precipitate from the solution and deposit onto the metal electrode. This process is known as ion precipitation, resulting in a positively charged metal electrode. When the dissolution and precipitation of ions reach dynamic equilibrium, a double layer is formed between the metal electrode and the solution, resulting in a potential difference known as the electrode potential of the corresponding electrode.

For different metals, their metal reactivity and ability to gain electrons vary, thereby

showing different electrode potentials. If different metal electrodes are immersed in the same electrolyte, the potential difference between the two electrodes can be measured with a voltmeter. In the case of the Zn/Ag/DNO sensor discussed in this manuscript, a hydrogel is used as a solid-state electrolyte to connect the two electrodes. The hydrogel contains a certain amount of Ca^{2+} , Cl^- , and abundant water molecules, which are chemically lower energy states for both Zn and Ag. As a result, both Zn and Ag electrodes exhibit a tendency for metal ions to diffuse into the hydrogel and show their respective electrode potentials. The measurement of OCV necessitates no external power source and the current in the circuit is close to zero, allowing the sensor to self-power.

We have supplemented the explanation in the revised manuscript. As shown in page 11:

“It is widely recognized that metal lattices comprise of systematically arranged metal ions and freely mobile electrons. When a metal electrode is submerged in water, the strongly polar water molecules tend to attract metal ions located within the metal lattices, thereby weakening the bond between certain metal ions and others present in the metal structure. The metal ions with weakened bonds then dissolve into the water, leading to a negative charge on the metal electrode due to loss of metal ions. Once the dissolution and precipitation of metal ions reach a dynamic equilibrium, the metal electrode demonstrates a stable electrode potential. Different metal electrodes display distinctive electrode potentials in the electrolyte, owing to their unique properties. Based on this principle, a self-powered H_2S sensing device with a steady OCV is manufactured utilizing two different metals as electrodes...When the sensors are used

for gas detection, the OCV can show a tight correlation with the H₂S concentration. As shown in Supplementary Fig. S11, the OCV of the Zn/Ag/DNH dropped sharply when exposed to H₂S and then recovered in N₂ gas, which is fully capable of being used for H₂S sensing.”

Supplementary Figure S11. Dynamic change in the Open-circuit voltage (OCV) of the device with Zn-DNH-Ag structure to H₂S gas with reduced concentration from 4 to 0.8 ppm.

4. Line 98 of the main text, expressions are no longer appropriate. The authors write “Nevertheless, research on hydrogel-based gas sensors is relatively infrequent, and hydrogel-based H₂S sensors have not even been invented so far.” In fact, Wang et al. invented the Cu-PAN hydrogel H₂S sensor. (Sensors & Actuators: B. Chemical 376 (2023) 132968; <https://doi.org/10.1016/j.snb.2022.132968>)

Response:

Thanks for the valuable comments from reviewer. We have revised the expressions in

the manuscript.

As shown in page 5: “Nevertheless, research on hydrogel-based gas sensors remains relatively infrequent, and the development of hydrogel-based H₂S sensors has been extremely limited thus far.”

5. The response value of the sensor was greatly influenced by the change of humidity and temperature, how to remove all the distractions?

Response:

Thanks for the valuable comments from reviewer. Note that the response value of the sensor is indeed influenced by variations in humidity and temperature. To eliminate the interference of humidity, a hydrophobic and breathable Ecoflex membrane was employed to encapsulate the sensor, effectively isolating it from environmental water molecules (Supplementary Fig. S23a). Notably, in Fig. 4a, the electrode was encapsulated using the same Ecoflex material to prevent direct gas contact with the electrode-hydrogel interface. However, the Ecoflex membrane used in this case was designed to be breathable, owing to its fabrication process involving spin-coating at a speed of 800 rpm. Consequently, the resulting membrane exhibited a relatively thin thickness. Moreover, the molecular gaps within the Ecoflex silicone matrix are comparatively larger than those of gases such as H₂S, N₂, and O₂, thereby enabling the permeation of small gas molecules through the membrane. In contrast, the Ecoflex configuration utilized in Fig. 4a was in a bulk form, featuring a thicker membrane that impedes molecular diffusion and effectively shields the electrode.

Herein, we conducted experiments to assess the response of the Zn/Ag/DNO sensor, which was wrapped with an Ecoflex membrane, to 0.8 ppm H₂S under varying RH conditions (Supplementary Fig. S23b). Remarkably, the sensor exhibits consistent response levels across the range of humidity conditions, with no significant decrease observed at the lower RH level of 37%. Furthermore, a notable observation was the distinct resistance of the encapsulated sensor to humidity interference when compared to the non-encapsulated Zn/Ag/DNO sensor, thus substantiating the feasibility and efficacy of this approach.

To address the issue of temperature interference, we can incorporate a temperature sensor to measure the ambient temperature, and subsequently calibrate the sensor's response based on the response curve obtained across varying temperatures.

We have supplemented discussion in the revised manuscript. As shown in page 17-18:

“And the influence of humidity on the response value can be further eliminated by encapsulation with hydrophobic and breathable elastomeric membranes (Supplementary Fig. S23)... To eliminate the interference of temperature, a temperature sensor can be employed to accurately measure the ambient temperature, and the gas sensor can subsequently be calibrated based on the response curve obtained at various temperatures.”

Supplementary Figure S23. Elimination of humidity interference on response. a Schematic diagram of Zn/Ag/DNO sensor with Ecoflex membrane encapsulation. **b** Response of the encapsulated sensor to 0.8 ppm H₂S under varying RH conditions (37%, 58%, 80%). Inset is a physical picture of the Zn/Ag/DNO sensor with Ecoflex membrane encapsulation.

6. A rough estimate from the response recovery curves of the sensors shows that the response/recovery times are all slow, but the authors do not mention this. Furthermore, the theoretical LOD of 0.73 ppb H₂S for the gas sensor is not obtained from the specific test curves, which can't be convincing enough.

Response:

Thanks for the valuable comments from reviewer. A segment of the dynamic response curve of Zn/Ag/Gly1h-DNO shown in Fig. 2c was extracted to analyse the response/recovery time at a specific concentration. The “t₉₀” refers to the time it takes for the response change to reach 90% of its peak value. As shown in Supplementary Fig. S17a, the Zn/Ag/Gly1h-DNO sensor displays a response time of 125.5 s and a

recovery time of 723.5 s to 2.4 ppm H₂S. Response/recovery time for other concentrations of H₂S are shown in Supplementary Fig. S17b. It can be seen that the sensor has shorter response/recovery time for higher concentrations of H₂S, and longer response/recovery time for lower concentrations. For instance, when the concentration drops from 4 ppm to 0.8 ppm, the response time increases from 96.4 s to 193.8 s. The response/recovery speed of the Zn/Ag/Gly1h-DNO sensor is not considered fast in comparison, which is a shortcoming of the sensor. Future research can explore increasing the chemical adsorption and desorption rates of gases on the electrodes to improve the sensor's response and recovery speed. The encouraging news is that the response time to low concentration of H₂S is within 200 s, making it capable of performing tasks such as halitosis detection and meat quality monitoring.

It is a pity that the lowest achievable concentration of H₂S under the current experimental conditions is limited to 20 ppb, which prevents obtaining an experimental test curve for lower concentrations. The theoretical LOD is calculated based on its definition and serves as indicative value to highlight the sensor's potential². It is worth noting that these calculated results are commonly employed as references for detection limits in many remarkable studies, such as *Advanced Functional Materials*, 2300046 (2023), *ACS Sensors* 7, 1183-1193 (2022), and *ACS Appl. Mater. Interfaces* 13, 46507-46517 (2021). In our case, the sensor shows an average response of 2.55 mV to 20 ppb H₂S. Using the methodology described in the literature, the sensor's noise level is calculated to be 0.006214 mV. By leveraging high-precision testing equipment and advanced algorithms, the attainment of a LOD as low as 0.73 ppb becomes a highly

feasible prospect.

We have supplemented the analysis of response/recovery time in the revised manuscript and supplementary information. As shown in page 15: “Nevertheless, the sensor exhibits a moderate response speed, which can be considered as an area for improvement (Supplementary Fig. S17).”

Supplementary Figure S17. Response/recovery time of Zn/Ag/DNO sensor. a

Analysis of response/recovery time to 2.4 ppm H₂S. The curve is a segment extracted from Fig. 2c. b Response/recovery time of sensor to varying concentrations of H₂S.

2. Shrivastava, A., Gupta, V. Methods for the determination of limit of detection and limit of quantitation of the analytical methods. *Chron. Young Sci.* **2**, 21-25 (2011).

7. Only one measurement of the sensing characteristics of the sensor at different humidities, temperatures and strain were reported for each concentration, please provide mean value and standard deviation in order to evaluate sensor repeatability.

Response:

Thanks for the valuable comments from reviewer. We have repeated the sensing performance characterization of the sensor under different humidity, temperature, and

strain with new samples, as shown in Fig. 3. The mean value and standard deviation are obtained from three consecutive gradient concentration tests. For the sake of rigor, we have replaced the previous data, which exhibited contingency, with supplementary data accompanied by error bars. Therefore, the corresponding data and discussion have been modified.

As shown in page 17-19: “With the increase of RH, the sensor responsivity increased gradually, from 13.6 mV at 37% RH to 55.7 mV at 80% RH for 0.8 ppm H₂S (Fig. 3b). The sensor exhibits greater response at 80% RH, and this can be explained that more H₂S molecules could be adsorbed and further reacted on the wetted electrode due to its hydrophilic nature, resulting in a larger response to the same H₂S concentration... And the influence of humidity on the response value can be further eliminated by encapsulation with hydrophobic and breathable elastomeric membranes (Supplementary Fig. S23). Then, the operating temperature range of the sensor was investigated. We measured the H₂S sensing performance of the Zn/Ag/DNO sensor at different temperatures from -20 to 40 °C (Fig. 3c-d), a temperature range that covers a large part of daily life. The gas sensing performance of the sensor at -20°C and 40°C exhibits similar characteristics, with sensitivities of 7.8 mV/ppm and 6.3 mV/ppm, and detection limits of 3.37 ppb and 3.98 ppb, respectively. Despite the reduced response compared to RT, it retains the capability to detect H₂S, thus satisfying the demands for detecting H₂S leakage in certain challenging operational conditions. To eliminate the interference of temperature, a temperature sensor can be employed to accurately measure the ambient temperature, and the gas sensor can subsequently be calibrated

based on the response curve obtained at various temperatures.

...It can be found that the responsiveness of the sensor is enhanced under tensile strain(Fig. 3f, g). The difference in response between the stretched and original states of the sensor can be attributed to changes in the interface, which can be addressed by further refining the structural design and implementing appropriate encapsulation techniques. The Zn/Ag/DNO sensor's ability to function effectively under strain makes it an ideal candidate for wearable electronics.”

Fig. 3 Sensing performance under variable environmental conditions and scenarios. a Dynamic responses of the Zn/Ag/DNO sensor to H₂S gas with reduced concentration from 4 to 0.8 ppm under different RH. b Response versus concentration curves of the Zn/Ag/DNO sensor at different RH. c Dynamic responses of the Zn/Ag/DNO sensor to H₂S gas with reduced concentration from 4 to 0.8 ppm under

different temperatures. **d** Response versus concentration curves of the Zn/Ag/DNO sensor under different temperatures. **e** Dynamic responses of the Zn/Ag/DNO sensor to H₂S gas with reduced concentration from 4 to 0.8 ppm under different external strains. **f** Response versus concentration curves of the Zn/Ag/DNO sensor under different external strains. **g** Sensitivities of the Zn/Ag/DNO sensor versus external strains. Inset is the photograph of the sensor being stretched to 100% strain. **h** Dynamic responses of Zn/Ag/DNO sensor to H₂S gas in air or N₂ background with reduced concentration from 4 to 0.8 ppm. **i** Sensitivities and theoretical LOD of Zn/Ag/DNO sensor to H₂S in air or N₂ background.

Reviewer #2 (Remarks to the Author):

In this manuscript, the authors reported a flexible and self-powered sensing device based on the structure of galvanic cells for monitoring of H₂S. The OCV is measured without an external power supply. Moreover, a water-rich and ion-conducting PAM/CA hydrogel was introduced as the solid-state electrolyte for improving the flexibility and stretchability of the system. Overall, the method demonstrated shows a decent prospect of real-time and remote monitoring of H₂S. I think this work will attract readers of Nature Communications, below are some detailed concerns before the manuscript can be published. Thus, I recommend this work to be published on Nature Communications after major revision.

1. It is pointed out that the sensor is self-powered, and there should be more corresponding descriptions and a physical picture should be added.

Response:

Thanks for the valuable suggestion. We have supplemented the description of self-powering and a physical picture in the revised manuscript.

As shown in page 11: “It is widely recognized that metal lattices comprise of systematically arranged metal ions and freely mobile electrons. When a metal electrode is submerged in water, the strongly polar water molecules tend to attract metal ions located within the metal lattices, thereby weakening the bond between certain metal ions and others present in the metal structure. The metal ions with weakened bonds then dissolve into the water, leading to a negative charge on the metal electrode due to loss of metal ions. Once the dissolution and precipitation of metal ions reach a dynamic equilibrium, the metal electrode demonstrates a stable electrode potential. Different metal electrodes display distinctive electrode potentials in the electrolyte, owing to their unique properties. Based on this principle, a self-powered H₂S sensing device with a steady OCV is manufactured utilizing two different metals as electrodes. For the convenience of presentation, the fabricated devices are declared as “metal 1/metal 2/electrolyte” (e.g., Zn/Ag/DNH), where metal 1 connects the negative electrode and metal 2 connects the positive electrode of test instrument, respectively (Supplementary Fig. S9).”

Supplementary Figure S9. A physical picture of the self-powered sensor (Zn/Ag/DNH).

2. SEM diagrams of the DNH should be given to show its microstructure.

Response:

Thanks for the valuable suggestion. To investigate the microstructural morphology of the DNH, the freeze-drying method was employed to remove water while preserving the interior structures of the hydrogel sample. Then, a layer of gold was sputtered onto the sample to make it conductive. The microstructure of DNH was observed by SEM (Supplementary Fig. S1). The microstructure of DNH was consistent with previous reports (Macromolecular Materials and Engineering 304, 1900227 (2019)), indicating that the hydrogel was successfully prepared. Corresponding discussion and diagram have been supplemented in the revised manuscript and supplementary information.

As shown in page 7: “From the scanning electron microscope (SEM) image of the freeze-dried DNH, the polymer components in the hydrogel appeared as a uniform interpenetrating porous structure in which water was filled (Supplementary Fig. S1), enabling easy mass transfer as in liquid electrolytes.”

Supplementary Figure S1. Scanning electron microscopy (SEM) image of double network (DN) hydrogel. The scale bar is 10 μm .

3. The authors chose PAM and CA to synthesize DNH. How about some other skeleton materials? Considering the mechanical properties and optical transparency, there should be many other candidates.

Response:

Thanks for the valuable comments from reviewer. It is worth noting that the sensor fabrication strategy in this paper is generally applicable to other skeleton materials. For example, we have utilized PAM/Chitosan (CS), PAM/Carrageenan (Carr), and PVA/Gly DNHs to develop sensors for the detection of H_2S gas, as shown in Fig. R1. The observed stimulus responsiveness to H_2S for each DNH indicates the

generalizability of the strategy, which may be of value in developing a wide range of sensing applications based on other types of skeleton materials. Further research can focus on the influence of the internal composition of the hydrogel on the gas-sensing performance and the search for materials with the best comprehensive properties.

Figure R1. Dynamic OCV change of sensors made from different DNHs to H₂S gas with varying concentrations. All sensors use Zn and Ag as electrodes. **a** PAM/CS; **b** PAM/Carr; **c** PVA/Gly.

4. In Supplementary Fig. S1, it would be more convincing that using videos instead of photographs to show how the hydrogel changes in tensile strains.

Response:

Thanks for the valuable suggestion. We have supplemented the corresponding video in the Supplementary files. And corresponding description have been supplemented in the revised manuscript.

As shown in page 7-8: “As shown in Supplementary Fig. S2 and Supplementary Movie 1, the hydrogel was able to be easily stretched up to 400% strain and remained intact and undamaged due to the complementary mechanical properties of the two polymer networks involved.”

Supplementary Figure S2. Photographs of pristine DN hydrogel (DNH) at 0% and 400% tensile strains.

5. For comparison, the long-term test of pristine hydrogel should be added in Supplementary Figure S12.

Response:

Thanks for the valuable suggestion. We have supplemented the long-term test of pristine hydrogel and the result has been added in Supplementary Figure S14. Minor revisions have been made to the corresponding section of the manuscript.

As shown in page 13-14: “During the long-term test (96 h), the OCV of the both Zn/Ag/DNH and Zn/Ag/DNO sensor remained basically stable (Supplementary Fig. S14), showing the stability of the devices.”

Supplementary Figure S14. OCVs of **Zn/Ag/DNH and Zn/Ag/DNO** sensors were intermittently recorded over 96 h, showing the stability of the OCV.

6. The authors stated that the developed sensor has excellent selectivity, but failed to give the reason. A detailed explanation is highly suggested, e.g. the adsorption energies between different gases and electrode metal based on DFT calculation.

Response:

Thanks for the valuable comments from reviewer. The excellent selectivity is attributed to the sulfurophilic nature of Ag and the promoting effect of H₂S on the dissolution of metal ions from the Ag electrode. Indeed, the electrode potential of the metal depends on the dissolution of metal ions in the electrolyte, as explained in detail in the supplement on page 11 of the revised manuscript, where we provided additional insights into the self-powering mechanism. Thus, the response of the sensor relies on the gas's influence on the dissolution of metal ions. Taking H₂S as an example, the metal electrode adsorbs H₂S, which promotes the dissolution of metal ions on the electrode.

The dissolved metal ions form a double layer at the electrode-hydrogel interface, causing a decrease in electrode potential due to the loss of metal cations. Other gases have a minor influence on the dissolution of metal ions on the electrode, resulting in either a small response or no response from the sensor. This can be verified by examining the electrochemical impedance spectroscopy (EIS) of the sensor under different gas atmospheres (Supplementary Fig. S22). We conducted EIS tests on the sensor in pure N₂, 4 ppm H₂S, 40000 ppm O₂, 4 ppm NO, and 1000 ppm ethanol, respectively, which could reflect the constant phase angle element (CPE) and double-layer capacitance at the electrode-hydrogel interface. After fitting the data, the CPE-T value for the sensor in the presence of H₂S was determined to be 33.8 μF, higher than that under other gas atmospheres. This indicates that the double layer under an H₂S atmosphere accumulates more charge, which originates from the dissolution of metal ions on the electrode. The theoretical calculations based on DFT serve as a supportive evidence indicating that the adsorption energy of H₂S on Ag falls within the range of weak chemisorption. However, we believe that the experimental results carry more persuasive weight in this regard.

Corresponding discussion has been supplemented in the revised manuscript. As shown in page 16: “And this could be attributed to the sulfurophilic nature of Ag and the promoting effect of H₂S on the dissolution of metal ions from the Ag electrode, which was further validated by the electrochemical impedance spectroscopy (EIS) of the sensor under different gas atmospheres (Supplementary Fig. S22).”

Supplementary Figure S22. Electrochemical impedance spectroscopy (EIS) analysis of the Zn/Ag/DNO sensor under different gas atmospheres. a Impedance spectra; b CPE-T values obtained by fitting. The insert is an equivalent circuit diagram of the Zn/Ag/DNO sensor.

7. Reproducibility is the key parameter in evaluating sensor performance, and therefore it should be explored as well.

Response:

Thanks for the valuable comments from reviewer. To explore reproducibility, two additional Zn/Ag/DNO samples were prepared and conducted H_2S gas sensing tests. Combining the existing data, the three Zn/Ag/DNO samples showed a similar response to H_2S , demonstrating the excellent reproducibility of the sensor (Fig. R2). We conducted a linear fit using the average response data, resulting in a sensitivity of 23.7 mV/ppm. This value will be utilized as a substitute for the previously reported sensitivity value.

Revision has been made in the manuscript, as shown in page 14: “For **three** constructed

Zn/Ag/DNO sensors, their average response versus H₂S concentration curve was shown in Fig. 2d according to its dynamic response curve towards different H₂S concentrations ranging from 4 to 0.8 ppm. The results show that there is a good linear relationship between the response and the H₂S concentration, which is beneficial for the practical resolution of the H₂S concentration. Besides, the small standard deviations in the response of the three Zn/Ag/DNO samples indicate the excellent reproducibility. Based on the corresponding linear fitting curve, the sensitivity of the Zn/Ag/ DNO sensor to H₂S was determined to be 23.7 mV/ppm, fully demonstrating its capability in detecting H₂S.”

Figure R2. **a** Dynamic responses of three Zn/Ag/DNO samples to H₂S gas with varying concentrations. **b** Average response (dots) of three Zn/Ag/DNO samples versus H₂S concentration and corresponding linear fitting line that revealed the sensitivity.

8. H₂S in several application scenarios, including halitosis diagnosis and meat spoilage identification. The comparison between the data obtained by the sensor and the laboratory equipment should be given.

Response:

Thanks for the valuable comments from reviewer. Due to the constraints of our experimental setup, we are unable to perform precise detection of H₂S using gas chromatography/mass spectrometry in the laboratory. Moreover, the current commercially available portable H₂S detectors have limited accuracy, which is insufficient for accurately measuring the concentration of H₂S in applications such as halitosis detection and meat quality monitoring. Nevertheless, existing literature indicates that human breath contains 694 compounds in respiratory analysis¹. Apart from H₂S, the exhaled breath of individuals with halitosis may also contain gases such as acetone, isoprene, methanethiol, and dimethyl sulfide. During the process of pork decomposition, bacteria break down proteins, resulting in the production of gases such as H₂S, methanethiol, dimethyl sulfide, NH₃, methane, and CO₂. In our previous manuscript, we tested gases including H₂S, NH₃, CO₂, and acetone. Now we have supplemented tests for methanol, isoprene, trichloromethane, and dimethyl sulfide, aiming to make the gas types covered in the selective testing as comprehensive as possible. As shown in Supplementary Fig. S21, the sensor exhibited a response of -10.69 mV to dimethyl sulfide, while showing no response to methanol, isoprene, and trichloromethane, demonstrating excellent selectivity as before. Based on the existing experimental results, it can be inferred that the response in applications such as halitosis detection and meat quality monitoring is primarily caused by H₂S.

We have supplemented the corresponding discussion in the revised manuscript. As shown in page 16: “In this case, we investigated the response of the Zn/Ag/DNO sensor to some possible interfering gases in the environment or in exhaled air at 25 °C, 58%

RH, including 40000 ppm O₂, 1.9 ppm NO, 200 ppm CO₂, 10 ppm NH₃, 1000 ppm ethanol, 1000 ppm isopropanol, 1000 ppm acetone, 1000 ppm methanol, 1000 ppm isoprene, 1000 ppm trichloromethane, and 1000 ppm dimethyl sulfide (Supplementary Fig. S21). Although 40000 ppm O₂ reduced the OCV of the sensor by 12.47 mV, it had little effect on the detection of H₂S when the O₂ concentration changed little. Furthermore, the sensor demonstrated a response of -10.69 mV to 1000 ppm dimethyl sulfide, implying its preference for sulfur-containing gases. Except for O₂ and dimethyl sulfide, the gas sensor exhibited a negligible response of 0.13 mV to 1.9 ppm NO and imperceptible responses to other interfering chemicals, demonstrating the excellent selectivity (Fig. 2i).”

Supplementary Figure S21. Selectivity of Zn/Ag/DNO sensor. a Plot showing the process that controlled the periodic “on” and “off” of target gases to Zn/Ag/DNO sensor in (b-l). **b-l** OCV of the Zn/Ag/DNO sensor in the cyclic detection of **b** 40000 ppm O₂, **c** 1.9 ppm NO, **d** 200 ppm CO₂, **e** 10 ppm NH₃, **f** 1000 ppm ethanol, **g** 1000 ppm

isopropanol, **h** 1000 ppm acetone, **i** 1000 ppm methanol, **j** 1000 ppm isoprene, **k** 1000 ppm trichloromethane, and **l** 1000 ppm dimethyl sulfide.

1. Van den Velde, S., van Steenberghe, D., Van Hee, P., Quiryne, M. Detection of odorous compounds in breath. *J. Dent. Res.* **88**, 285-289 (2009).

9. The error bars should be added such as in Fig. 2D and Supplementary Figure S4.

Response:

Thanks for the valuable comments from reviewer. We have repeated the corresponding experiments and added error bars in Fig. 2d, Supplementary Figure S5, Fig. 3b, Fig. 3d, and Fig. 3f. The previous data has been replaced with supplementary data accompanied by error bars.

As shown in page 14: “For **three** constructed Zn/Ag/DNO **sensors**, **their average** response versus H₂S concentration curve was shown in Fig. 2d according to its dynamic response curve towards different H₂S concentrations ranging from 4 to 0.8 ppm. The results show that there is a good linear relationship between the response and the H₂S concentration, which is beneficial for the practical resolution of the H₂S concentration. **Besides, the small standard deviations in the response of the three Zn/Ag/DNO samples indicate the excellent reproducibility.** Based on the corresponding linear fitting curve, the sensitivity of the Zn/Ag/ DNO sensor to H₂S was determined to be **23.7** mV/ppm, fully demonstrating its capability in detecting H₂S.”

As shown in page 9: “Within 10 h, the mass loss of Gly1h-DNO, Gly2h-DNO and Gly4h-DNO was greatly reduced to **38.2%, 31.7%, and 18.9%**, respectively,

demonstrating the enhanced moisturizing ability.”

As shown in page 17-19: “With the increase of RH, the sensor responsivity increased gradually, from 13.6 mV at 37% RH to 55.7 mV at 80% RH for 0.8 ppm H₂S (Fig. 3b).

The sensor exhibits greater response at 80% RH, and this can be explained that more H₂S molecules could be adsorbed and further reacted on the wetted electrode due to its hydrophilic nature, resulting in a larger response to the same H₂S concentration... And

the influence of humidity on the response value can be further eliminated by encapsulation with hydrophobic and breathable elastomeric membranes (Supplementary Fig. S23). Then, the operating temperature range of the sensor was

investigated. We measured the H₂S sensing performance of the Zn/Ag/DNO sensor at different temperatures from -20 to 40 °C (Fig. 3c-d), a temperature range that covers a large part of daily life. The gas sensing performance of the sensor at -20°C and 40°C

exhibits similar characteristics, with sensitivities of 7.8 mV/ppm and 6.3 mV/ppm, and detection limits of 3.37 ppb and 3.98 ppb, respectively. Despite the reduced response compared to RT, it retains the capability to detect H₂S, thus satisfying the demands for

detecting H₂S leakage in certain challenging operational conditions. To eliminate the interference of temperature, a temperature sensor can be employed to accurately measure the ambient temperature, and the gas sensor can subsequently be calibrated based on the response curve obtained at various temperatures.

...It can be found that the responsiveness of the sensor is enhanced under tensile strain (Fig. 3f, g). The difference in response between the stretched and original states of the sensor can be attributed to changes in the interface, which can be addressed by

further refining the structural design and implementing appropriate encapsulation techniques. The Zn/Ag/DNO sensor's ability to function effectively under strain makes it an ideal candidate for wearable electronics.”

Fig. 2d. Average response (dots) of three Zn/Ag/Gly1h-DNO samples versus H₂S concentration and corresponding linear fitting line that revealed the sensitivity.

Supplementary Figure S5. Relative weight change of pristine DNHs and DNOs when stored at 25°C, 40% RH for a series of time.

Fig. 3 Sensing performance under variable environmental conditions and scenarios. **a** Dynamic responses of the Zn/Ag/DNO sensor to H₂S gas with reduced concentration from 4 to 0.8 ppm under different RH. **b** Response versus concentration curves of the Zn/Ag/DNO sensor at different RH. **c** Dynamic responses of the Zn/Ag/DNO sensor to H₂S gas with reduced concentration from 4 to 0.8 ppm under different temperatures. **d** Response versus concentration curves of the Zn/Ag/DNO sensor under different temperatures. **e** Dynamic responses of the Zn/Ag/DNO sensor to H₂S gas with reduced concentration from 4 to 0.8 ppm under different external strains. **f** Response versus concentration curves of the Zn/Ag/DNO sensor under different external strains. **g** Sensitivities of the Zn/Ag/DNO sensor versus external strains. Inset is the photograph of the sensor being stretched to 100% strain. **h** Dynamic responses of Zn/Ag/DNO sensor to H₂S gas in air or N₂ background with reduced concentration from

4 to 0.8 ppm. i Sensitivities and theoretical LOD of Zn/Ag/DNO sensor to H₂S in air or N₂ background.

10. It should be “*V_{H₂S}*” instead of “V_{H₂S}”. Please note that all the variables should be displayed in italic form.

Response:

Thanks for the valuable suggestion. All variables have been italicized in the revised manuscript.

As shown in page 11: “The response here is defined as

$$Resp = \Delta V = V_{H_2S} - V_0 \quad (1)$$

where *V_{H₂S}* and *V₀* are the stabilized OCV in flowing H₂S and background gas, respectively.”

As shown in page 14: “...can be estimated from the slope of the response versus gas concentration curve:

$$S = \left| \frac{\Delta Resp}{\Delta C} \right| \quad (2)$$

As shown in page 20: “Note that the OCV we measured is expressed as

$$OCV = E_{Ag} - E_{Zn} \quad (3)$$

where *E_{Ag}* and *E_{Zn}* are the electrode potentials of Ag and Zn electrodes... which can also lead to a small decrease in *E_{Zn}* and a small increase in the final OCV.”

11. It is suggested that the English expression can be further polished by a language expert.

Response:

Thanks for the valuable suggestion. The English expression in Abstract, Introduction, and Discussion sections has been polished by a language expert.

As shown in page 2: “Timely and remote biomarker detection is highly desired in personalized medicine and health protection but presents great challenges in the devices reported so far. Here, we **present** a novel cost-effective, flexible and self-powered sensing device for H₂S biomarker analysis in various application scenarios based on the structure of galvanic cells. **The sensing mechanism is attributed to the change in electrode potential resulting from the chemical adsorption of gas molecules on the electrode surfaces.** Intrinsically stretchable organohydrogels are used as solid-state electrolytes to enable stable and long-term operation of devices under stretching deformation or in various environments. The resulting open-circuit sensing device exhibits high sensitivity, low detection limit, and excellent selectivity for H₂S. Its application in the non-invasive halitosis diagnosis and identification of meat spoilage is demonstrated, emerging great commercial value in portable medical electronics and food security. A wireless sensory system has also been developed for remote H₂S monitoring with the participation of Bluetooth and cloud technologies. **This work breaks through the shortcomings in the traditional chemiresistive sensors, offering a new direction and theoretical foundation for designing wearable sensors catering to other stimulus detection requirements.**”

As shown in page 3: “**Over the past few decades, biomarker detection has garnered considerable attention in the medical and health domain due to its potential for early**

disease diagnosis and personalized health monitoring, and it is expected to play a crucial role in the forthcoming era of Cloud Medical Treatment (CMT)... Besides, continuous monitoring of relatively high concentrations of H₂S is imperative due to its toxic nature. Prolonged exposure to H₂S levels exceeding 10 ppm can have severe detrimental effects on human health.”

As shown in page 27-28: “In summary, a novel flexible, room-temperature operable, and self-powered gas sensor was proposed for the first time based on the susceptibility of the electrode potential to the target gas, with the advantages of outstanding sensing performance and wide environmental adaptability. Given its unique attributes, this sensor could serve as an ideal candidate for wearable electronics. Comprised of two metal electrodes and a solid hydrogel electrolyte derived from a galvanic cell, the sensor exhibits a straightforward and cost-effective design, enabling arbitrary stretching and bending due to the intrinsic flexibility of electrolyte... Furthermore, a compact wireless sensing system is designed and developed, capable of wirelessly transmitting the detected signals to the user terminal through Bluetooth or cloud sharing technology, enabling real-time and remote H₂S monitoring. This work presents a pioneering concept and fundamental basis for the future development of low-cost, self-powered, and wearable gas sensors with high performance at RT. This significant advancement holds great promise in enhancing human health and safety, contributing to the continuous progress in the field of sensor technology.”

12. Several recently published work on room-temperature gas sensors are

recommended for citation. (e.g. Chemical Engineering Journal, 2022, 446: 136937; ACS Sensors, 2022, 7 (4): 1183-1193; Analytical Chemistry, 2020, 92 (16): 11277-11287).

Response:

Thanks for the valuable suggestion. The recommended works have been cited in the revised manuscript.

As shown in page 5: “One strategy to achieve flexible gas sensor is integrating sensing materials on soft substrates (*e.g.*, polyethylene terephthalate, polydimethylsiloxane, polyimide and Ecoflex)¹⁷⁻²².”

As shown in page 16: “Based on the sensitivity and the noise level, the theoretical LOD can be calculated as 0.79 ppb, which is enough to tell if the test subject has a bad breath problem or a tendency to halitosis (Supplementary Fig. S20, Table S1)⁴¹⁻⁴³.”

Reviewer #3:

In this manuscript, Wu and coworkers report a flexible and self-powered H₂S gas sensing device by adopting the structure of galvanic cells and using stretchable organohydrogels as solid-state electrolytes. The device shows superior gas sensing performance at room temperature and maintains its functionality at sub-zero temperatures or under mechanical deformations, which is unattainable by existing H₂S sensors. The demonstrations of halitosis diagnosis, meat freshness monitoring and real-time remote H₂S leakage alarm reveal huge application potential in human health and

safety protection. Additionally, this paper provides a novel strategy for the design of flexible, self-powered gas sensors and creates the necessary groundwork for subsequent research. In general, this work is very interesting and worthy of investigation, therefore I would like to recommend the publication in Nature Communications after addressing the following minor issues.

1. According to Supplementary Fig. S18 and Fig. 2i, both H₂S and O₂ decrease the open circuit voltage of the sensor. However, H₂S is a reducing gas while O₂ is an oxidizing gas. Is the adsorption site and effect of O₂ molecules on the sensor the same as H₂S? The effect of oxygen adsorption on the electrode potential deserves further investigation.

Response:

Thanks for the valuable comments from reviewer. We conducted response testing experiment on O₂ using the Zn/Ag/DNH sensor, as well as experiment with the Zn electrode shielded, as shown in Fig. R3. The results revealed that the OCV of the pristine Zn/Ag/DNH sensor decreased when O₂ was introduced, consistent with the results shown in Supplementary Fig. S21. Conversely, the OCV of the Zn-encapsulated sensor increased when O₂ was introduced. According to Equation 3, O₂ causes an increase in the electrode potential of Ag, as the electrode potential of Zn is not influenced by external gases when the Zn electrode is shielded. Combining this information with the fact that the OCV of the unencapsulated sensor decreases when O₂ is introduced, we can infer that in the absence of a shielded electrode, O₂ causes an increase in the electrode potential of Zn, and the increase is greater than that of Ag. Therefore, the adsorption sites of O₂ are at the Zn-hydrogel interface and the Ag-

hydrogel interface, which increase the electrode potentials of Zn and Ag, contrary to the effect of H₂S. The focus of this work is on H₂S, and the discussion on O₂ can be further explored in future study.

$$OCV = E_{Ag} - E_{Zn} \quad (3)$$

Figure R3. Selective shielding electrode experiment of Zn/Ag/DNH sensor to O₂. **a** Dynamic OCV of pristine Zn/Ag/DNH sensor to varying concentrations of O₂. **b** Dynamic OCV of Zn-encapsulated Zn/Ag/DNH sensor to varying concentrations of O₂.

2.The EDS spectrum is missing in Supplementary Fig. S23-24 and Fig. 4f. The author should add the EDS spectrum to quantitatively analyze the electrode composition changes before and after the experiment.

Response:

Thanks for the valuable comments from reviewer. We have added the EDS spectrum in the supplementary information, and the corresponding discussion has been supplemented in the revised manuscript.

As shown in page 22-23: “To validate this, the Zn/Ag/DNO sensor underwent an extended OCV test (96 h), and the surface morphology of both the Zn and Ag electrodes was examined using scanning electron microscopy (SEM) before and after the test

(Supplementary Fig. S26, Fig. 4f). Notably, no significant corrosion was observed on either electrode, confirming their robustness and stability. The elemental composition of the electrodes was quantitatively analyzed using EDS, and after 96 h, a slight increase in the amount of oxygen (O) was detected on the Zn electrode, which can be attributed to the natural oxidation of Zn in the humid environment (Supplementary Fig. S27). Whereas in the short-circuit (SC) state, only 8 h of continuous testing resulted in severe damage to the Zn electrode due to electrochemical reactions (Supplementary Fig. S28), making it unsuitable for long-term applications. EDS analysis also revealed a significant increase in the amount of O on the Zn electrode, indicating more pronounced corrosion of Zn (Supplementary Fig. S29).”

Supplementary Figure S27. EDS element analysis before and after keeping the Zn/Ag/DNO in OC state for 96 h. a Zn wire before test; b Zn wire after test; c Ag wire before test; d Ag wire after test.

Supplementary Figure S29. EDS element analysis before and after keeping the Zn/Ag/DNO in SC state for 8 h. a Zn wire before test; b Zn wire after test; c Ag wire before test; d Ag wire after test.

3. How to guarantee the uniformity of the developed device, which is very important for the commercialization of the device. In addition, the newly proposed sensing device is millimeter-sized, and whether it can be further miniaturized?

Response:

We are deeply grateful for the reviewer's valuable suggestions. In the actual commercialization process, in order to reduce the device-to-device variation, the most important thing is to maintain the consistency of the composition and structure of the sensitive material and the connection between the electrode and the sensitive material. In this work, we carefully control the proportions of the components in the hydrogel precursor solution to maintain consistency in the composition of the electrolyte. Furthermore, uniform-sized electrodes and the same number of winding turns are used

to ensure the consistency of the devices as much as possible. Since this work represents an initial exploration of this novel H₂S gas sensor, there may be some errors introduced due to manual handling. In future studies, standardized processes can be implemented to cut the hydrogel blocks, prepare electrodes, and connect electrode materials with hydrogel materials, ensuring device consistency.

Based on the sensing mechanism proposed in our study, the response of the sensor does not depend on the size of the hydrogel but rather on the appropriate electrode and interface. Therefore, the sensing device can be further miniaturized by reducing the volume of the hydrogel, for example, by fabricating the hydrogel into thin films or fiber structures. This would require the incorporation of standardized processes for electrode fabrication.

4. Normally, water-containing hydrogels cannot be used directly for XPS analysis because the vacuum of the cavity is not up to the test requirements. What treatment was given to the hydrogel samples in the article (Fig. 4c-d)? It is recommended that the preparation process of the hydrogel samples for XPS analysis be supplemented in the “Methods” section.

Response:

Thanks for the valuable comments from reviewer. The preparation process of hydrogel samples for XPS analysis is as follows: a Zn/Ag/DNH sensor was exposed to 2 ppm H₂S for 10 h. Subsequently, the hydrogel from the sensor was extracted, and hydrogel blocks were cut from the regions where they were wrapped around the Zn and Ag

electrodes. The hydrogel blocks were then placed in a drying oven at 110 °C for 2 h.

The obtained dried hydrogel blocks were subjected to XPS analysis.

It has been supplemented in the “Methods” section of the revised manuscript. As shown

in page 29: **Synthesis of Zn/Ag/DNH and Zn/Ag/DNO sensor.** A Zn/Ag/DNH sensor

was prepared by winding Zn wire and Ag wire at the two ends of a DN hydrogel block.

Unless otherwise specified, the Zn wire has a diameter of 0.3 mm and the Ag wire has

a diameter of 0.06 mm, with a winding count of 3 turns for both. By replacing the DN

hydrogel in the Zn/Ag/DNH sensor with a DN organohydrogel, a Zn/Ag/DNO sensor

can be obtained.

Synthesis of hydrogel samples for XPS analysis. A Zn/Ag/DNH sensor was exposed

to 2 ppm H₂S for 10 h. Subsequently, the hydrogel from the sensor was extracted, and

hydrogel blocks were cut from the regions where they were wrapped around the Zn and

Ag electrodes. The hydrogel blocks were then placed in a drying oven at 110 °C for 2

h. The obtained dried hydrogel blocks were subjected to XPS analysis.”

5. The result in Fig. 3b shows that the response of the sensor increases with the increase of relative humidity. While in the halitosis detection experiment (Fig. 5a), the humidity of dry air and the collected exhaled gas is significantly different. Does the difference in humidity between the two gases affect the experimental results? Please explain this in detail.

Response:

Thanks for the valuable comments from reviewer. The response value of the sensor is

indeed influenced by variations in humidity, and this issue can be addressed by encapsulating the sensor with hydrophobic and breathable elastomeric membranes. As supplemented in page 17: “And the influence of humidity on the response value can be further eliminated by encapsulation with hydrophobic and breathable elastomeric membranes (Supplementary Fig. S23).”

In the experimental setup for halitosis detection, we used saturated K_2SO_4 solution to humidify the air and minimize the humidity difference between the target gas and background air. The vapor relative humidity of the saturated K_2SO_4 solution is 98%, and at a flow rate of 100 sccm, the air can be sufficiently humidified. Additionally, the background air was kept consistent during three tests for the target gases, ensuring reliable experimental results.

We have supplemented the description of saturated K_2SO_4 solution in the revised manuscript. As shown in page 24: “In the device shown in Fig. 5a, simulated halitosis gas/exhaled gas and dry air were alternately delivered to the Zn/Ag/DNO sensor, and their responses were recorded. To achieve comparable humidity levels between the target gas and background gas, a saturated K_2SO_4 solution was employed to humidify the gases.”

Reviewers' Comments:

Reviewer #1:

Remarks to the Author:

The authors responded carefully my comments and revised the manuscript. Therefore, I would like to recommend acceptance of the manuscript.

Reviewer #2:

None

Reviewer #3:

Remarks to the Author:

I have no question now and I recommend the publication of this manuscript. We think that the readers of Nature Communications would be of interest to this manuscript.